# Comprehensive integrative analyses identify *GLT8D1* and *CSNK2B* as schizophrenia risk genes

Cui-Ping Yang[1], Xiaoyan Li[1], Yong Wu[1], Qiushuo Shen[1,2], Yong Zeng[3], Qiuxia Xiong[3], Mengping Wei[4], Chunhui Chen[5], Jiewei Liu[1], Yongxia Huo[1], Kaiqin Li[1], Gui Xue[5], Yong-Gang Yao[1,6], Chen Zhang[4], Ming Li [1,6], Yongbin Chen[1,7] & Xiong-Jian Luo [1,7]

Recent genome-wide association studies (GWAS) have identified multiple risk loci that show strong associations with schizophrenia. However, pinpointing the potential causal genes at the reported loci remains a major challenge. Here we identify candidate causal genes for schizophrenia using an integrative genomic approach. *Sherlock* integrative analysis shows that *ALMS1*, *GLT8D1*, and *CSNK2B* are schizophrenia risk genes, which are validated using independent brain expression quantitative trait loci (eQTL) data and integrative analysis method (*SMR*). Consistently, gene expression analysis in schizophrenia cases and controls further supports the potential role of these three genes in the pathogenesis of schizophrenia. Finally, we show that *GLT8D1* and *CSNK2B* knockdown promote the proliferation and inhibit the differentiation abilities of neural stem cells, and alter morphology and synaptic transmission of neurons. These convergent lines of evidence suggest that the *ALMS1*, *CSNK2B*, and *GLT8D1* genes may be involved in pathophysiology of schizophrenia.

[1] Key Laboratory of Animal Models and Human Disease Mechanisms of the Chinese Academy of Sciences and Yunnan Province, Kunming Institute of Zoology, Chinese Academy of Sciences, Kunming, Yunnan 650223, China. [2] Kunming College of Life Science, University of Chinese Academy of Sciences, Kunming, 650204, China. [3] Department of Psychiatry, The First Affiliated Hospital of Kunming Medical College, Kunming, Yunnan 650031, China. [4] State Key Laboratory of Membrane Biology, PKU-IDG/McGovern Institute for Brain Research, School of Life Sciences, Peking University, Beijing 100871, China. [5] State Key Laboratory of Cognitive Neuroscience and Learning, and IDG/McGovern Institute for Brain Research, Beijing Normal University, Beijing 100875, China. [6] CAS Center for Excellence in Brain Science and Intelligence Technology, Chinese Academy of Sciences, Shanghai 200031, China. [7] Center for Excellence in Animal Evolution and Genetics, Chinese Academy of Sciences, Kunming, Yunnna 650223, China. These authors contributed equally: Cui-Ping Yang, Xiaoyan Li, Yong Wu, and Qiushuo Shen. These authors are jointly supervised this work: Ming Li, Yongbin Chen, and Xiong-Jian Luo. Correspondence and requests for materials should be addressed to Y.C. (email: ybchen@mail.kiz.ac.cn) or to X.-J.L. (email: luoxiongjian@mail.kiz.ac.cn)

Schizophrenia (SCZ) is a severe mental disorder with a lifetime prevalence of ~ 0.5–1% across all human populations[1]. This disease is characterized by positive symptoms (i.e., delusions and hallucinations), negative symptoms (i.e., apathy, impaired motivation, and social withdrawal), and cognitive impairment (i.e., disorganized thoughts, impaired working memory, and executive function)[2]. As one of the most common mental diseases, SCZ has a high mortality and significant long-term morbidity[3]. Persons with SCZ have a high risk for suicide[4]. Approximately 30% of patients attempt suicide and eventually about 10% patients die from suicide[5]. Accordingly, the economic burden of SCZ is particularly great. For example, in 2002, about $62.7 billion was spent on SCZ in the United States[6].

Even though SCZ has a high prevalence and poses a severe risk to global health[1,2,7], currently there are no proven treatment options that can cure this disease completely. A major reason for this therapeutic stasis is that we know little about the etiology of SCZ. Multiple lines of evidence suggest that SCZ has both genetic and environmental components[8]. Family, twin, and adoption studies have demonstrated that SCZ has a strong genetic factor, with an estimated heritability around 0.80[9]. In fact, SCZ has the highest heritability among neuropsychiatric disorders[10], indicating the pivotal role of the inherited risk variants in SCZ. To uncover the genetic underpinnings of SCZ, multiple large-scale genetic studies have been performed[11–20]. Despite the fact that multiple promising candidate genes have been identified by genetic linkage and association studies[21,22], the vast majority of the heritability of SCZ remains unexplained. The application of genome-wide association studies (GWAS) to SCZ greatly accelerated the discovery of novel risk variants. In 2008, O'Donovan et al.[23] first reported the association of *ZNF804A* with SCZ in a GWAS. Subsequently, multiple GWAS have been performed in different ethnic populations and a number of genome-wide significant variants/loci have been identified[11–20]. Recently, the Schizophrenia Working Group of the Psychiatric Genomics Consortium (PGC) reported the association results of the largest SCZ GWAS so far (PGC2 release)[20]. PGC identified 108 independent SCZ-associated genetic loci (83 are newly reported in their paper), which greatly expands the genome-wide significant genetic variants (or loci) and provides novel biological insights for SCZ.

To date, more than 100 independent risk variants have been identified by GWAS[19,20]. Nevertheless, how the reported variants confer risk of SCZ remains elusive, as most of the risk variants are located in non-coding genomic regions[20]. To identify genes whose expression levels are affected by SCZ risk variants and to explore the potential role of these genes in SCZ pathogenesis, we performed comprehensive integrative analyses by integrating data from different layers, including genetic associations from large-scale GWAS (PGC2)[20], brain expression quantitative trait loci (eQTL), protein–protein interaction (PPI), co-expression, spatio-temporal expression pattern, differential gene expression in SCZ cases and controls, association with human brain structure, and in vitro functional studies. Our integrative analyses showed that *ALMS1*, *CSNK2B*, and *GLT8D1* genes may represent promising risk genes for SCZ.

## Results

**Sherlock integrative analysis prioritizes 10 SCZ risk genes.** The overview of the study design is provided in Supplementary Fig. 1. To identify genes whose expression level change may confer risk of SCZ, we systematically integrated genetic associations (a total of 9,444,230 single-nucleotide polymorphisms (SNPs)) from the largest GWAS of SCZ so far (including 35,476 cases and 46,839 controls)[20] with brain eQTL from 193 normal human brain samples[24]. The statistical inference was conducted by *Sherlock* integrative analysis[25], a statistical method to detect disease-associated genes by matching patterns of eQTL and GWAS. Through using the *Sherlock* Bayesian statistical inference method to match the "signature" of genes from brain eQTL with patterns of association in GWAS, we identified 10 top SCZ susceptibility genes whose expression level may affect SCZ risk at $P < 5.0 \times 10^{-6}$ (Bonferroni corrected $P < 0.05$) (Table 1 and Supplementary Table 1). These genes are *LOC376138*, *KDELR1*, *ALMS1*, *GLT8D1*, *ZNF323*, *CSNK2B*, *SLITRK5*, *LOC375768*, *TBC1D15*, and *ZNF446*. For each gene, at least one SNP (usually several SNPs) showed significant association with the expression of this gene and SCZ risk simultaneously, implying that these SNPs are promising functional candidates with potential regulation effect.

**Validation of *GLT8D1* and *CSNK2B* as SCZ risk genes.** *Sherlock* integrative analysis had yielded a list of promising candidates whose expression change may contribute to SCZ risk (the first layer prioritization, 10 genes) (Table 1). To further verify our findings, we replicated the results of *Sherlock* integrative analysis using an independent brain eQTL data set (from Fromer et al.[26],

## Table 1 *Sherlock* integrative analysis identifies *ALMS1*, *GLT8D1*, and *CSNK2B* as schizophrenia risk genes

| Gene symbol | LBF[a] | P-value[b] | Supporting SNP[c] (cis or trans) | $P_{eQTL}$[d] | $P_{GWAS}$[e] | Corrected $P$[f] |
|---|---|---|---|---|---|---|
| *LOC376138* | 9.38 | $1.11 \times 10^{-6}$ | rs4898494 (*trans*) | $3.08 \times 10^{-17}$ | $7.04 \times 10^{-6}$ | < 0.05 |
| *KDELR1* | 8.82 | $1.11 \times 10^{-6}$ | rs4735158 (*trans*) | $8.04 \times 10^{-6}$ | $3.33 \times 10^{-9}$ | < 0.05 |
| *ALMS1* | 7.63 | $1.11 \times 10^{-6}$ | rs6753344 (*cis*) | $1.37 \times 10^{-9}$ | $1.77 \times 10^{-6}$ | < 0.05 |
| *GLT8D1* | 7.10 | $1.11 \times 10^{-6}$ | rs6795646 (*cis*) | $2.50 \times 10^{-8}$ | $7.15 \times 10^{-9}$ | < 0.05 |
| *ZNF323* | 7.05 | $1.11 \times 10^{-6}$ | rs1150709 (*cis*) | $2.05 \times 10^{-8}$ | $7.05 \times 10^{-10}$ | < 0.05 |
| *CSNK2B* | 6.70 | $1.11 \times 10^{-6}$ | rs2075800 (*cis*) | $5.39 \times 10^{-6}$ | $3.63 \times 10^{-11}$ | < 0.05 |
| *SLITRK5* | 6.16 | $2.21 \times 10^{-6}$ | rs1265883 (*cis*) | $7.49 \times 10^{-6}$ | $5.04 \times 10^{-5}$ | < 0.05 |
| *LOC375768* | 5.93 | $2.21 \times 10^{-6}$ | rs9853627 (*trans*) | $4.85 \times 10^{-7}$ | $1.09 \times 10^{-6}$ | < 0.05 |
| *TBC1D15* | 5.65 | $4.42 \times 10^{-6}$ | rs6582065 (*cis*) | $4.96 \times 10^{-6}$ | $5.12 \times 10^{-5}$ | < 0.05 |
| *ZNF446* | 5.64 | $4.42 \times 10^{-6}$ | rs7012579 (*trans*) | $4.38 \times 10^{-7}$ | $1.51 \times 10^{-5}$ | < 0.05 |

[a]LBF (the logarithm of the Bayes factor for the gene) is a critical indicator to assess whether a specific gene is associated with schizophrenia through integrating the evidence from GWAS and brain eQTL. The larger the LBF, the higher probability that the gene is associated with schizophrenia. For example, a LBF of 5.64 means that the interested gene is more likely (281 times, (exp(5.64) = 281)) to be associated with schizophrenia than the null hypothesis (no association). For a given gene, *Sherlock* first identifies all SNPs that are associated with the expression of this gene (these SNPs were called eSNPs). *Sherlock* then scores each eSNP based on the association significance between the eSNP and schizophrenia. A positive score would be assigned to the eSNP if this eSNP is also significantly associated with schizophrenia. A negative score would be assigned to the eSNP if this eSNP is not associated with schizophrenia. Disease associations found only in the GWAS do not alter the score. For a given gene, individual LBF for each eSNP of this gene was calculated and the sum of the LBFs was used as the final LBF score of this gene
[b]P-value derived from *Sherlock* statistical inferences
[c]Only the SNP with the highest LBF is listed
[d]P-value from expression quantitative trait analysis[33]
[e]P-value from GWAS of schizophrenia[20]
[f]P-value was corrected by Bonferroni multiple testing correction

**Table 2 Validation of *ALMS1*, *GLT8D1*, and *CSNK2B* as schizophrenia risk genes using independent brain eQTL data set (*Sherlock* integrative analysis)**

| Gene symbol | LBF | *P*-value | Supporting SNP (*cis* or *trans*) | $P_{eQTL}$[a] | $P_{GWAS}$ | Corrected *P* |
|---|---|---|---|---|---|---|
| *ALMS1* | 6.34 | $3.25 \times 10^{-6}$ | rs56145559 (*cis*) | $2.11 \times 10^{-16}$ | $8.42 \times 10^{-8}$ | < 0.05 |
| *GLT8D1* | 7.23 | $4.07 \times 10^{-7}$ | rs7620039 (*cis*) | $2.93 \times 10^{-9}$ | $3.47 \times 10^{-8}$ | < 0.01 |
| *CSNK2B* | 6.15 | $5.69 \times 10^{-6}$ | rs144024892 (*cis*) | $8.49 \times 10^{-12}$ | $4.39 \times 10^{-11}$ | < 0.052 |

[a]*Sherlock* integrative analysis was performed using brain eQTL from Fromer et al.[26] (brain tissues (DLPFC) were collected from a total of 467 individuals)

**Table 3 Validation of *ALMS1* and *GLT8D1* as schizophrenia risk genes using independent integrative analysis method (*SMR*)[68]**

| Gene symbol | Top SNP (*cis* or *trans*) | $P_{eQTL}$[a] | $P_{GWAS}$ | $P_{SMR}$[b] | Corrected $P_{SMR}$ |
|---|---|---|---|---|---|
| *ALMS1* | rs7607892 (*cis*) | $1.54 \times 10^{-9}$ | $3.19 \times 10^{-6}$ | $2.22 \times 10^{-4}$ | < 0.05 |
| *GLT8D1* | rs1961958 (*cis*) | $6.41 \times 10^{-27}$ | $2.41 \times 10^{-7}$ | $3.45 \times 10^{-6}$ | < 0.05 |

[a,b]*SMR* integrative analysis was performed using schizophrenia GWAS from the PGC[20] and brain eQTL from Myers et al.[24]

$N = 467$). Compared with the brain eQTL data from Myers et al.[24] (which was used in the discovery stage of *Sherlock* integrative analysis in this study), brain eQTL data from Fromer et al.[26] (we called this brain eQTL as replication brain eQTL data set) has a larger sample size. Thus, replication of *Sherlock* analysis results using this brain eQTL data set will provide robust evidence to support our original findings. Genome-wide SNP associations from the PGC[20] and independent brain eQTL data (from Fromer et al.)[26] were used as input for *Sherlock* integrative analysis, as described in the Methods section and the previous studies[26,27]. *Sherlock* integrative analysis using independent brain eQTL data successfully validated *ALMS1*, *CSNK2B*, and *GLT8D1* as SCZ risk genes (corrected *P* for *ALMS1* and *GLT8D1* < 0.05, corrected *P* for *CSNK2B* = 0.052, Table 2). We further performed *Sherlock* analysis using a third brain eQTL (from GTEx[28], brain tissues of 92 subjects were used). Again, we found that *GLT8D1* is significantly associated with SCZ ($P = 4.8 \times 10^{-4}$). *Sherlock* did not detect *ALMS1* and *CSNK2B* in the GTEx data set. Successful replication of *ALMS1*, *CSNK2B*, and *GLT8D1* as SCZ risk genes using independent and larger brain eQTL data strongly suggests that the three genes may be authentic SCZ susceptibility genes. Collectively, these consistent results provided further evidence that support *ALMS1*, *CSNK2B*, and *GLT8D1* as promising risk genes for SCZ.

***SMR* analysis supports *ALMS1* and *GLT8D1* as SCZ risk genes**. To further validate our results, we utilized a different integrative analysis method (i.e., summary data-based Mendelian randomization, *SMR*)[29] to identify SCZ risk genes through integrating SCZ GWAS from PGC[20] and brain eQTL from Myers et al.[24]. Intriguingly, *SMR* analysis showed that *GLT8D1* and *ALMS1* are significantly associated with SCZ (Table 3), providing further evidence that supports these two genes as authentic risk genes for SCZ. It should be noted that *Sherlock* and *SMR* use different strategies to identify SCZ risk genes. In this study, same SCZ GWAS[20] and brain eQTL data[24] were used as input for *Sherlock* and *SMR*. Thus, the identification of *GLT8D1* and *ALMS1* by independent integrative analysis methods (i.e., *Sherlock* and *SMR*) provided further evidence to support *GLT8D1* and *ALMS1* as authentic risk genes for SCZ.

**The identified risk genes are expressed in human brain**. Despite decades of intensifying research, the pathogenesis of SCZ remains largely unknown. Accumulating evidence suggests that SCZ is

likely a neurodevelopmental disorder[30]. Consistent with the neurodevelopmental hypothesis, recent studies have shown that several SCZ risk genes (including *DISC1* and *RELN*) have an important role in brain development through regulating proliferation and migration of neuronal progenitors[31,32]. If the SCZ candidate genes identified by *Sherlock* integrative analysis are involved in brain development, they may be expressed in developing the human brain. We therefore explored the expression pattern of the identified candidate genes in diverse human tissues using the Gene Enrichment Profiler[33]. We found that 7 of the 10 top candidate genes identified by *Sherlock* were widely expressed in human brain regions (Supplementary Fig. 2). Interestingly, we noticed that these genes are also extensively expressed in immune-related cells (Supplementary Fig. 3). Further spatio-temporal expression pattern analysis in the developing prefrontal cortex showed that the mRNA expression levels of most of the genes are higher at early developmental stage (i.e., embryonic and fetal stages) compared with childhood and adulthood stages (Fig. 1). Permutation test showed that the observed expression pattern is unlikely due to chance effect ($P = 4 \times 10^{-3}$), suggesting that SCZ risk genes identified by *Sherlock* have higher expression level at early developmental stage than later stage. These spatio-temporal expression pattern results suggest that the genes identified by *Sherlock* integrative analysis may have a role in the human brain.

**PPI among risk genes identified by Sherlock and GWAS**. Accumulating evidence implies that proteins encoded by SCZ susceptibility genes are preferably interacted and form a highly interconnected network[34–36]. To explore whether proteins encoded by genes identified by *Sherlock* physically interacted with proteins encoded by GWAS-significant SCZ risk genes (Supplementary Table 2), we performed PPI analysis using interactions from the GeneMANIA PPI database[37]. We found that proteins encoded by genes identified by *Sherlock* significantly interacted with proteins encoded by GWAS-significant SCZ risk genes (permutation $P = 0.045$) (Supplementary Fig. 4). Among these genes, CSNK2B has the most number of interactions with proteins encoded by GWAS-significant SCZ susceptibility genes. Recently, Rossin et al.[38] developed a method (named DAPPLE) to prioritize disease-associated genes through using PPI data (based on the "guilt by association" principle). We therefore prioritized the top 10 genes (Table 1) identified by *Sherlock* using the DAPPLE program[38]. The results suggested that *ALMS1* ($P = 0.071$), *CSNK2B* ($P = 0.063$), and *TBC1D15* ($P = 0.019$) may

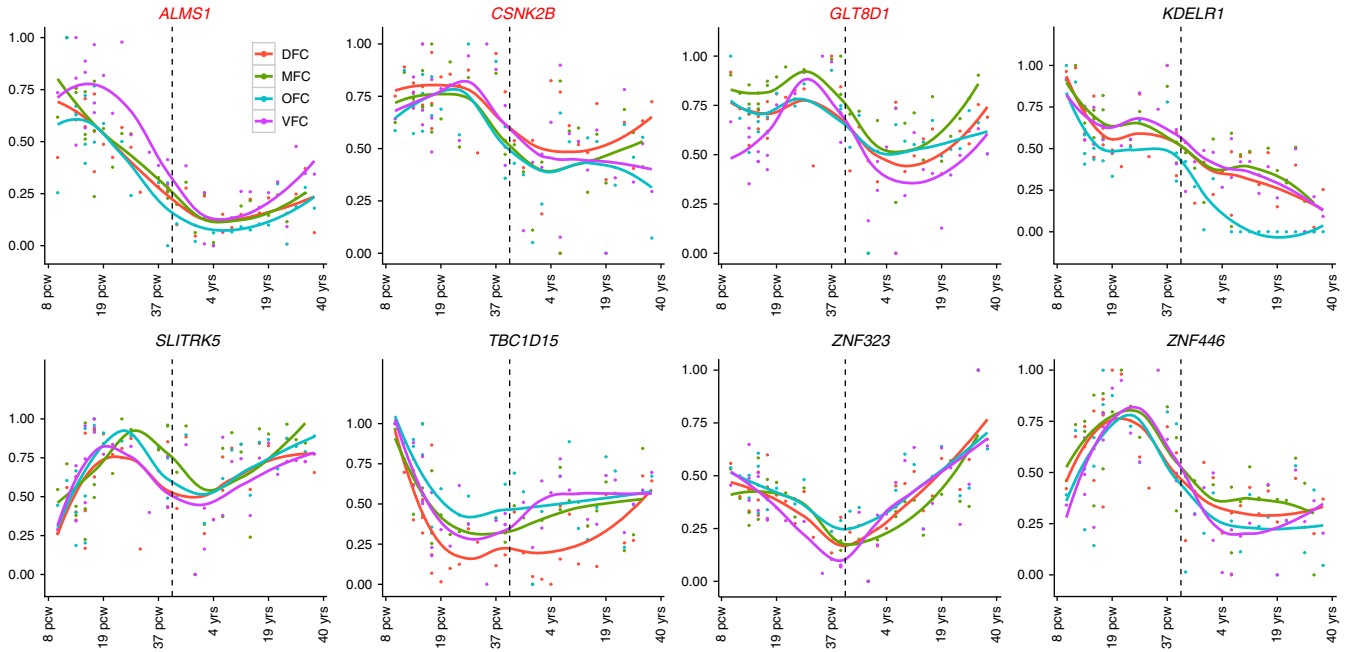

**Fig. 1** Expression patterns of the top predicted risk genes (by *Sherlock*) in human frontal cortex. Expression level of top predicted risk genes across the entire developing stages (from 8 post-conception weeks (pcw) to 40 years (yrs)) were depicted in the frontal cortex, which was divided into four regions: dorsolateral prefrontal cortex (DFC), medial prefrontal cortex (MFC), orbital prefrontal cortex (OFC), and ventrolateral prefrontal cortex (VFC). Of note, most of the genes showed higher expression level at embryonic and fetal stages (8 pcw to 4 mons) compared with childhood and adulthood stages (8 yrs to 40 yrs), suggesting these genes may have a role in neurodevelopment. The expression data (from frontal cortex of 42 human subjects) were extracted from the BrainSpan[73]

represent promising risk genes for SCZ. However, as the *P*-values of *ALMS1* and *CSNK2B* did not reach significance level, the PPI results of *ALMS1* and *CSNK2B* were not used for scoring in following prioritization (Supplementary Fig. 1).

**Genes identified by Sherlock and GWAS are co-expressed.** Previous study has shown that genes harboring damaging de novo mutations in SCZ are significantly co-expressed in human brain and form a highly interconnected co-expression network[34]. To explore whether genes identified by *Sherlock* are co-expressed with SCZ risk genes identified by GWAS (Supplementary Table 2) in the human brain, we performed co-expression analysis using the expression data from the BrainSpan (http://www.brainspan.org/). As described in a previous study[39], we divided the brain tissues into four main clusters, including the V1C-STC cluster (V1C, ITC, IPC, A1C, and STC), PFC-MSC cluster (M1C, S1C, ventrolateral prefrontal cortex (VFC), medial prefrontal cortex (MFC), dorsolateral prefrontal cortex (DFC), and orbital prefrontal cortex (OFC)), STR-AMY cluster (STR, HIP, and AMY), and MD-CBC cluster (MD and CBC). If the absolute value of the Pearson's correlation coefficient of a gene pair is > 0.8, this gene pair was connected in the co-expression network[34]. We found that genes identified by *Sherlock* are significantly co-expressed with genome-wide significant SCZ risk genes in the V1C-STC cluster ($P = 4 \times 10^{-3}$), STR-AMY cluster ($P = 1.2 \times 10^{-2}$), and MD-CBC cluster ($P < 1.0 \times 10^{-3}$) (Supplementary Fig. 5b-d). In the PFC-MSC cluster, genes identified by *Sherlock* also showed a trend of significant co-expression ($P = 8.8 \times 10^{-2}$) with GWAS-significant genes (Supplementary Fig. 5a). Of note, *ALMS1* has the most connections (i.e., co-expressed with SCZ susceptibility genes identified by GWAS) in the PFC-MSC cluster (Supplementary Fig. 5a) and STR-AMY cluster (Supplementary Fig. 5c). In the V1C-STC cluster, *ALMS1* and *CSNK2B* have the most number of connections with GWAS-

significant genes (Supplementary Fig. 5d). *SLITRK5* has the most number of connections (Supplementary Fig. 5b) in the MD-CBC cluster. These co-expression results further suggest that *ALMS1* and *CSNK2B* may represent promising risk genes for SCZ.

**Dysregulation of GLT8D1 and CSNK2B in SCZ cases.** *Sherlock* integrative analysis suggested that genes listed in Table 1 may represent promising SCZ risk genes whose expression level change may contribute to SCZ risk. In addition, we also prioritized the top 10 genes identified by *Sherlock* using the DAPPLE (utilizing PPI data) and co-expression network analyses, and the results suggest that *AMLS1* and *CSNK2B* may represent plausible causal genes for SCZ (as these two genes are frequently interacted or co-expressed with genome-wide significant SCZ risk genes). Of note, *Sherlock* infers disease-associated genes under the assumption that the expression level of the candidate genes are altered in patients. Thus, if the genes identified by *Sherlock* are true risk genes, the expression of these genes should be dysregulated in SCZ. To further validate the results of *Sherlock* integrative analysis, we examined the expression of the distilled risk genes in SCZ cases and healthy controls. We found that *AMLS1* is significantly upregulated in the hippocampus ($P = 0.006$) and striatum ($P = 0.041$) of SCZ cases compared with controls (Fig. 2a) in GSE53987 data set[40]. In the prefrontal cortex, *AMLS1* showed a trend of upregulation ($P = 0.056$). *GLT8D1* and *CSNK2B* are significantly downregulated in the hippocampus of SCZ cases ($P = 0.006$ and 0.005, respectively) (Fig. 2b, c). Interestingly, in GSE12649 data set[41], *AMLS1* is also significantly upregulated (35 SCZ cases and 34 controls) in the prefrontal cortex of SCZ cases compared with controls ($P = 0.019$) (Supplementary Fig. 6). *GLT8D1* and *CSNK2B* did not show significant change in GSE12649 data set[41]. Three genes (*LOC376138*, *ZNF323*, and *LOC375768*) were not included in GSE53987[40] and GSE12649[41] data sets, whereas other genes (*KEELR1*, *SLITRK5*,

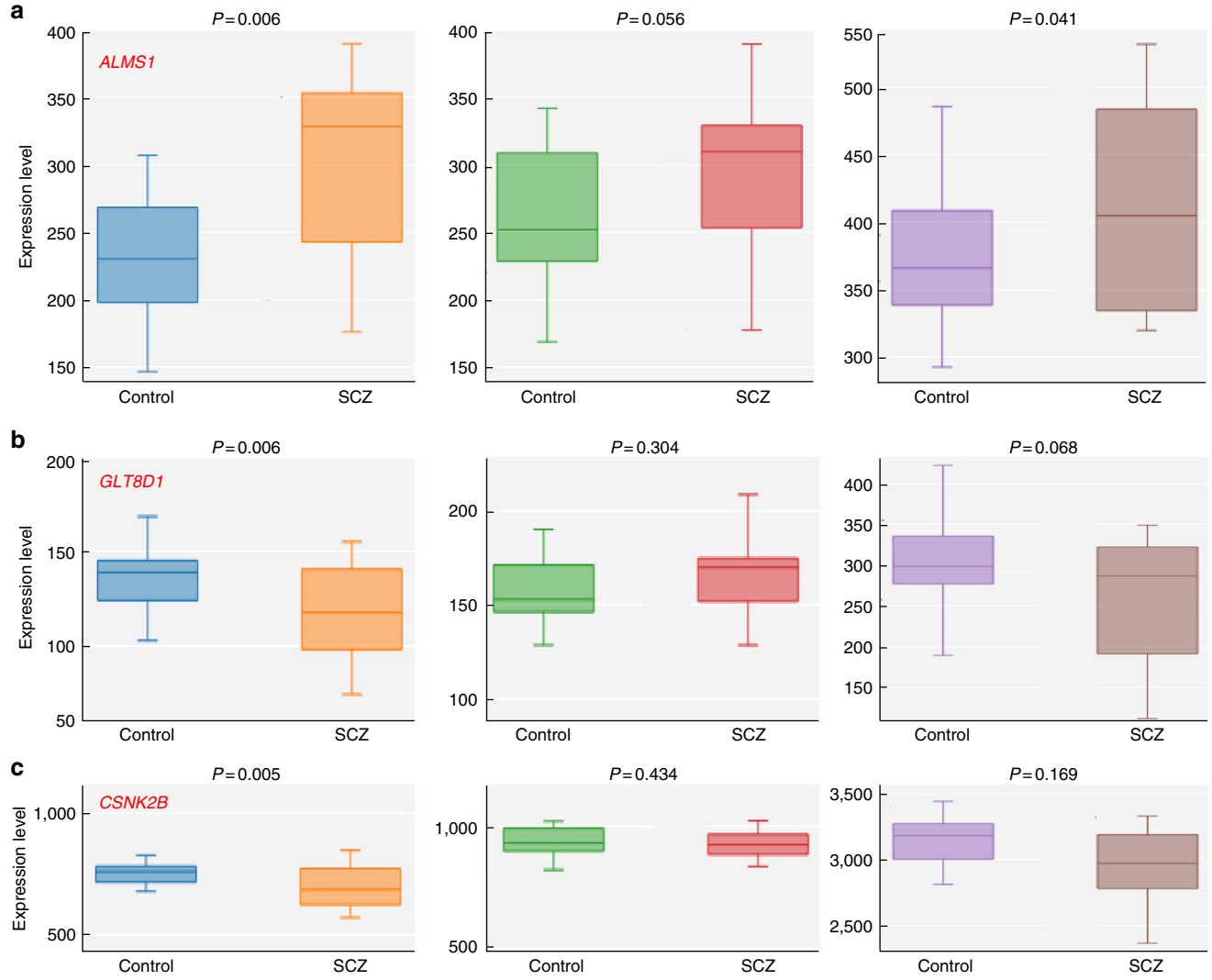

**Fig. 2** Dysregulation of *ALMS1*, *GLT8D1*, and *CSNK2B* in schizophrenia cases vs. healthy controls. **a** mRNA expression of *AMLS1* is significantly upregulated in the hippocampus and striatum (P = 0.006 and 0.041, respectively) of schizophrenia cases (SCZ) compared with healthy subjects (Control) (GSE53987 expression data set[40], 19 schizophrenia cases, and 19 controls). In the prefrontal cortex, *AMLS1* also showed a trend of up-regulation (P = 0.056). **b**, **c** Expression of *GLT8D1* and *CSNK2B* is significantly decreased in the hippocampus of schizophrenia cases compared with controls. Left panel, hippocampus, middle panel, prefrontal cortex, right panel, and striatum. Student's *t*-test was used to compare whether the expression of *ALMS1*, *GLT8D1*, and *CSNK2B* was significantly different in schizophrenia cases (n = 19) and controls (n = 19)

*TBC1D15*, and *ZNF446*) did not show a significant expression change in SCZ cases.

To further verify the dysregulation of *AMLS1* in SCZ, we explored the mRNA expression of *AMLS1* in the Stanley neuropathology consortium integrative database[42]. Again, we found that *AMLS1* is significantly upregulated in the frontal cortex of SCZ cases compared with controls in two independent data sets[42] (Supplementary Fig. 7). Of note, we found that *ALMS1* is consistently upregulated in SCZ cases in several independent data sets[40–42]. Although *GLT8D1* and *CSNK2B* did not show significant change in GSE12649, a recent study[43] showed that *GLT8D1* is significantly downregulated in SCZ cases, which is consistent with our observation in GES53987[40]. These expression data suggest that dysregulation of *ALMS1*, *CSNK2B*, and *GLT8D1* may have a role in SCZ.

**ALMS1 is associated with hippocampal structure**. Our comprehensive integrative analyses, including *Sherlock* integrative analysis, network-based prioritization (PPI and co-expression),

differential expression, and *SMR* analyses supported that *ALMS1*, *CSNK2B*, and *GLT8D1* are risk genes for SCZ. Of note, previous studies have shown that SCZ risk variants are also associated with the hippocampal structure in healthy subjects[44,45]. To explore whether the newly identified SCZ-associated eSNPs of *AMLS1*, *GLT8D1*, and *CSNK2B* (identified by *Sherlock* and *SMR* integrative analysis, Table 1) are associated with hippocampal structure in normal individuals, we examined the association between eSNPs of *AMLS1*, *GLT8D1*, *CSNK2B* (Table 1), and hippocampal structure using the data from the Enhanced Neuro Imaging Genetics through Meta-Analysis (ENIGMA) consortium[46]. We found that SCZ-associated eSNPs of *ALMS1* (i.e., rs6753344, rs56145559, and rs7607892) are associated with hippocampal volume (P = 0.024, 0.026, and 0.019, respectively) (Supplementary Tables 3, 4, and 5). SCZ-associated eSNPs of *GLT8D1* and *CSNK2B* are not associated with hippocampal structure (Supplementary Tables 3, 4, and 5). These results suggest that the *ALMS1* gene may have a role in brain development, further supporting the involvement of this gene in SCZ.

**GLT8D1 and CSNK2B are associated with cognitive function**. We further investigated the association between eSNPs of *ALMS1*, *CSNK2B*, *GLT8D1*, and working memory in healthy individuals. We found that SCZ-associated eSNPs of *AMLS1* gene (rs6753344, rs56145559, and rs7607892, Table 1) are close to monomorphic in Han Chinese (Supplementary Table 6). We thus focused on *CSNK2B* and *GLT8D1*. Interestingly, we found that eSNPs of *CSNK2B* (rs2075800) and *GLT8D1* (rs6795646) (Table 1) are significantly associated with working memory ($P = 0.048$ and 0.049, respectively) (Supplementary Table 7) in healthy individuals. In addition to working memory, we found that eSNP of *CSNK2B* (rs2075800) is also significantly associated with long-term memory ($P = 0.022$). Interestingly, we noticed that the risk allele of SNP rs2075800 (T allele) is associated with poorer working memory (Supplementary Table 7). These results suggest that eSNPs of *CSNK2B* and *GLT8D1* are associated with cognitive function. We noted that the association is moderate, which is consistent with previous findings, i.e., cognitive function is affected by numerous genetic variants and the effect of single genetic variant is small[47]. These results provide further evidence that supports *CSNK2B* and *GLT8D1* may be authentic risk genes for SCZ.

**GLT8D1 and CSNK2B regulate the proliferation of mouse NSCs**. To further explore the functional roles of *ALMS1*, *CSNK2B*, and *GLT8D1* in the pathogenesis of SCZ, we sought to characterize the functions of these three genes in central nervous system (CNS). Previous studies showed that *ALMS1* is expressed in the human brain and mutations of *ALMS1* causes Alström syndrome[48,49], a disorder characterized by childhood obesity, and vision and hearing loss. *CSNK2B* (also known *CK2B*) encodes the β-regulatory subunit of casein kinase II, an important protein kinase that has pivotal role in regulating metabolism, signal transduction, cell proliferation, and survival[50].

In contrast to *ALMS1* and *CSNK2B*, the function of *GLT8D1* (encodes a member of the glycosyltransferase family) is largely unknown. Considering that *ALMS1* is upregulated whereas the expression of *GLT8D1* and *CSNK2B* are downregulated in SCZ cases compared with controls (Fig. 2), we thus decided to deplete the expression of *GLT8D1* and *CSNK2B* in the CNS and characterize their functional roles. Mouse neural stem cells (NSCs) (derived from embryonic day 14.5) were transfected with small hairpin RNAs (shRNAs) targeting *GLT8D1* and *CSNK2B* (Methods), respectively, and the proliferation, self-renewal, and differentiation abilities of NSCs were evaluated. We first assessed the knockdown efficiency of the designed shRNAs. Both real-time PCR (*GAPDH* was used an the internal control) and western blotting showed that the shRNAs reduced the mRNA and protein expressions of *GLT8D1* and *CSNK2B* significantly compared with respective controls (Fig. 3a, b and Supplementary Fig. 8). Interestingly, the proliferation ratios (5-bromo-2′-deoxyuridine (BrdU)-positive NSCs) were significantly increased in *GLT8D1* and *CSNK2B* knockdown NSCs compared with controls, respectively (Fig. 3c, d). We then analyzed the stemness marker genes (such as *Klf4*, *Sox2*, *Nanog*, and *Nestin*) expression pattern in *GLT8D1* and *CSNK2B* knockdown NSCs compared with control NSCs, and found that these marker genes were unanimously upregulated by depleting *GLT8D1* and *CSNK2B* (Fig. 3e). These results indicated that *GLT8D1* and *CSNK2B* have important roles in regulating the proliferation and stemness of NSCs.

**GLT8D1 and CSNK2B regulate the differentiation of mouse NSCs**. We further investigated the functional roles of *GLT8D1*

and *CSNK2B* in NSCs differentiation and found that the ratio of Tuj1 (a marker of newly generated immature postmitotic neurons), microtubule-associated protein 2 (MAP2, a marker of mature neurons), glial fibrillary acidic protein (GFAP, a marker of glia cells), and O4 (a marker of oligodendritic cells)-positive cells, as well as the mRNA expression levels of the key NSCs differentiation regulators, such as *Neurod1*, *Gfap*, *oligo1*, and *oligo2* were significantly decreased in *GLT8D1* and *CSNK2B* knockdown groups compared with controls (Figs. 4 and 5a, b, and Supplementary Fig. 9), indicating that the differentiation abilities of NSCs were impaired by *GLT8D1* and *CSNK2B* knockdown. Consistently, similar phenotypes were observed in *GLT8D1* knockdown NSCs isolated and cultured from P7 dentate gyrus (Supplementary Fig. 10 and 11). Interestingly, we noticed that mRNA level of *β-catenin* and *Axin2* (two important regulators in Wnt signaling pathway) showed significant upregulation in *GLT8D1-* and *CSNK2B-* knockdown NSCs (*GAPDH* was used as the internal control; Supplementary Fig. 12 and 13). However, protein level of β-catenin did not show significant change. Taken together, these data indicated that *GLT8D1* and *CSNK2B* have important roles in neurodevelopment through regulating the proliferation and differentiation of NSCs.

**GLT8D1 and CSNK2B regulate the morphology of neurons**. It has been demonstrated that dysfunction of SCZ risk genes led to morphologic defects of neurons[51,52]. To examine whether *GLT8D1* and *CSNK2B* have similar effects, we compared the morphology of neurons differentiated from *GLT8D1* and *CSNK2B* knockdown NSCs with that in control neurons differentiated from NSCs expressing scramble control shRNA. Our data showed that the dendritic length, number of dendrites, and branch points were significantly reduced in neurons after *GLT8D1* and *CSNK2B* knockdown compared with controls, respectively (Fig. 5c, d, e, f).

**GLT8D1 and CSNK2B regulate synaptic transmission**. In addition to morphologic defects, dysfunction of synaptic transmission has also been well characterized in SCZ[53–55]. We found that knockdown of *GLT8D1* and *CSNK2B* in neurons significantly altered the frequency of miniature excitatory postsynaptic currents (mEPSC) and amplitude of miniature inhibitory postsynaptic currents (mIPSC) (Fig. 6), implying that *GLT8D1* and *CSNK2B* are required for normal synaptic transmission. Taken together, these results suggested that *GLT8D1* and *CSNK2B* have pivotal roles in regulating dendritic development and modulating synaptic transmission, which provided further evidences supporting that *GLT8D1* and *CSNK2B* are promising SCZ risk genes.

**Integrative genomic method identifies three SCZ risk genes**. To combine the lines of evidence from different analyses, we used the convergent functional genomics (CFG)[56] method to prioritize the identified genes (see Methods). Through integrating the multiple independent lines of evidence from different analyses (Supplementary Fig. 1), including Sherlock integrative analysis, PPI, co-expression, differential gene expression, associations with brain structure and cognitive, and in vitro functional assays, we found that *ALMS1*, *CSNK2B*, and *GLT8D1* have the highest final scores (Supplementary Fig. 1), strongly suggesting these three genes are promising candidates for SCZ risk.

## Discussion

Hitherto, more than 100 SCZ-associated loci have been identified by GWAS study[20]. Nevertheless, how to pinpoint the causal variants and genes remains a major challenge. Owing to

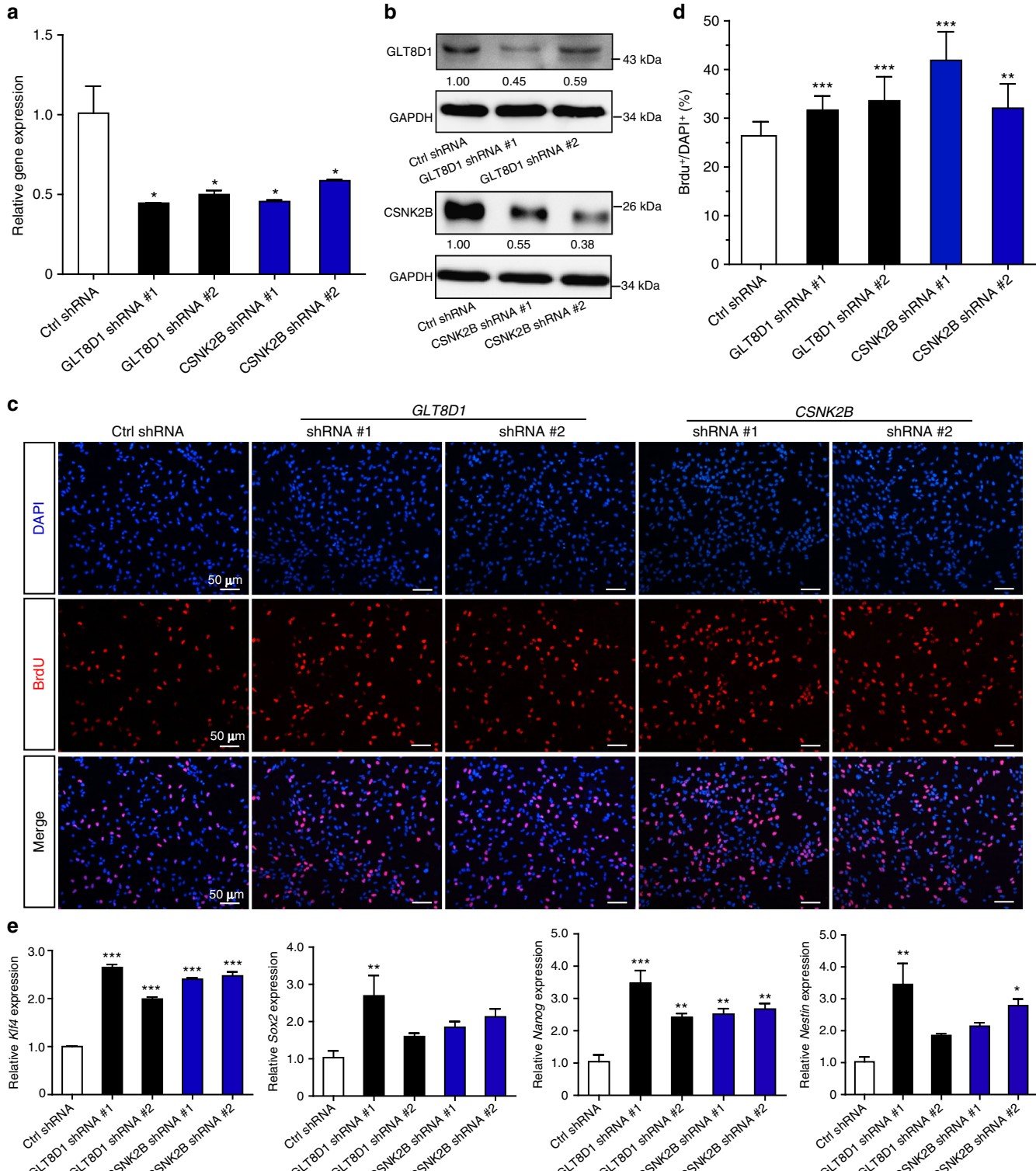

**Fig. 3** Knockdown of *GLT8D1* and *CSNK2B* promote NSCs proliferation and upregulate the expression of the stemness marker genes. **a** Real-time PCR showed that the shRNAs significantly reduced the mRNA expression level of *GLT8D1* and *CSNK2B* in NSCs. **b** Western blotting showed that protein level of GLT8D1 and CSNK2B was also significantly reduced. **c**, **d** Knockdown of *GLT8D1* and *CSNK2B* promote the proliferation ability of NSCs by BrdU incorporation assay. **d** The quantification data for **c**. **e** Knockdown of *GLT8D1* and *CSNK2B* increased the expression of stemness (*Sox2*, *Klf4*, *Nanog*, and *Nestin*) marker genes significantly. *GAPDH* was used an internal control. *$P < 0.05$, **$P < 0.01$, ***$P < 0.001$, two-tailed Student's $t$-test. Data represent mean ± SD from three independent biological replicates

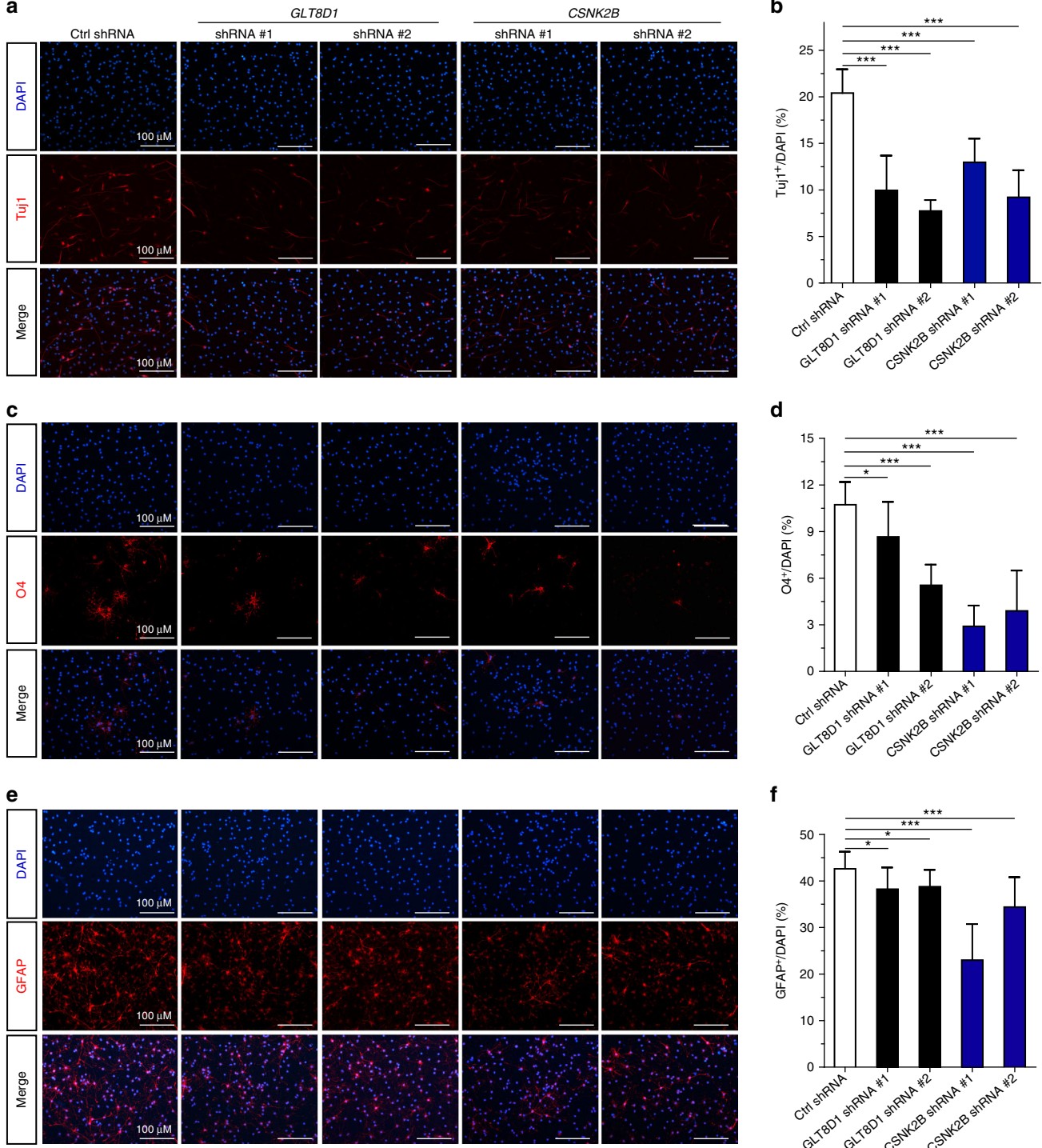

**Fig. 4** Knockdown of *GLT8D1* and *CSNK2B* inhibit the differentiation ability of NSCs. **a**, **b** Knockdown of *GLT8D1* and *CSNK2B* inhibited the differentiation of NSCs into neurons. **a** Representative immunostainings for Tuj1 (a marker for newly generated immature neurons) in indicated cells. **b** The quantification data for **a**. **c**, **d** Knockdown of *GLT8D1* and *CSNK2B* significantly inhibited the differentiation of NSCs into oligodendrocytes. **c** Representative immunostainings for O4 (a marker for oligodendrocytes) in indicated cells. **d** The quantification data for **d**. **e**, **f** Knockdown of *GLT8D1* and *CSNK2B* inhibited the differentiation of NSCs into astrocytes significantly. **e** Representative immunostainings for GFAP (a marker for astrocytes) in indicated cells. **f** The quantification data for **e**. Scale bar: 100 μM. *$P < 0.05$, **$P < 0.01$, ***$P < 0.001$, two-tailed Student's *t*-test. Data represent mean ± SD from three independent biological assays

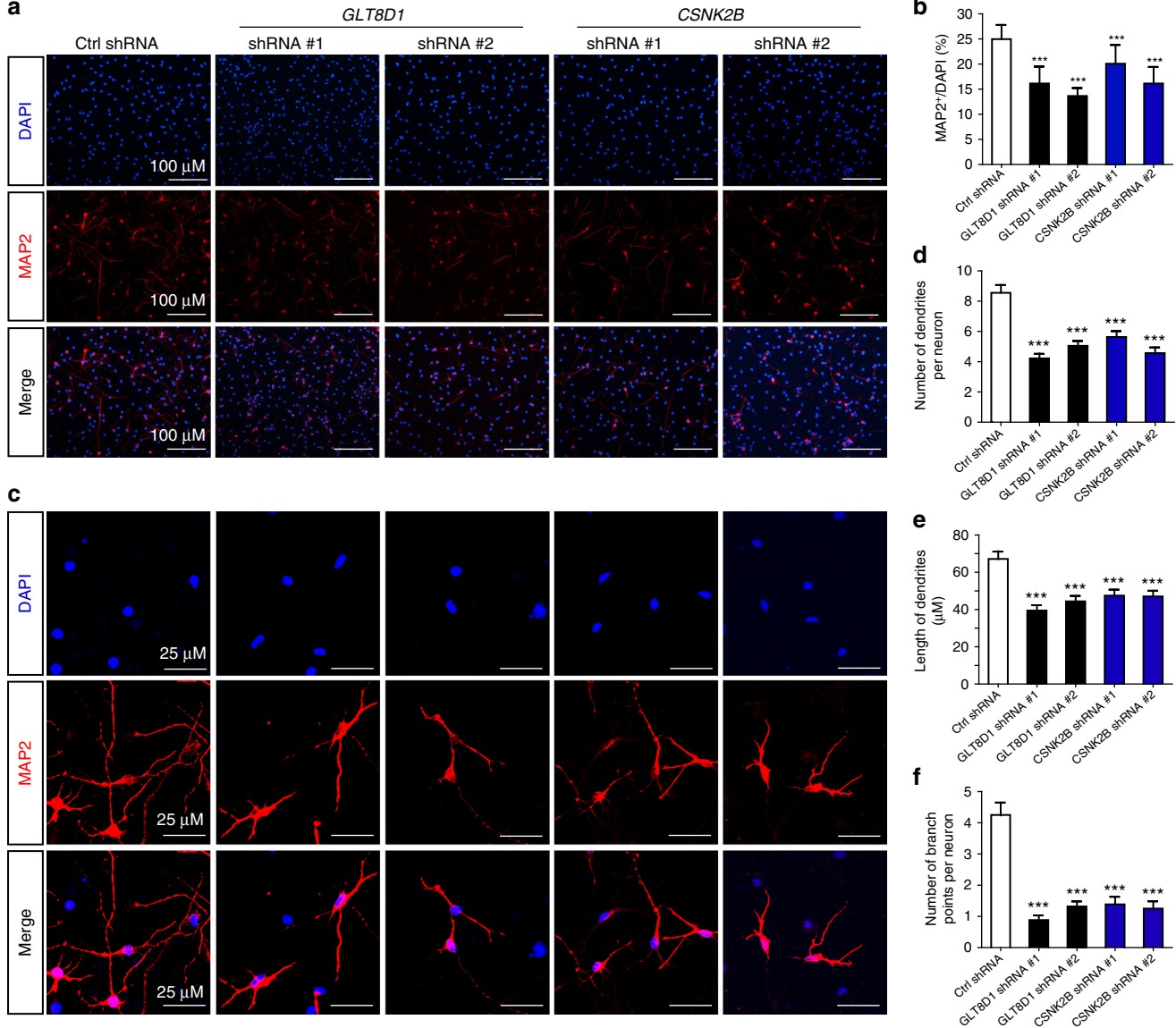

**Fig. 5** Knockdown of *GLT8D1* and *CSNK2B* lead to defects in dendritic morphology. **a** Representative immunostainings for MAP2 (a marker for neurons) in indicated cells. **b** Quantification data for **a**. **c–f** Knockdown of *GLT8D1* and *CSNK2B* led to defects in dendritic growth. Compared with controls, the number of dendrites, the dendritic length, and branch points were significantly reduced in neurons differentiated from NSCs that stably expressing shRNAs targeting *GLT8D1* and *CSNK2B*. A total of 60 neurons were measured for the control and experimental groups. Data in **b** represent mean ± SD. Data in **d–f** represent mean ± SEM. ***$P < 0.001$, two-tailed Student's *t*-test

the linkage disequilibrium, the reported risk loci usually contain many highly linked genetic variants that showed similar association significance. For each identified risk locus, it is a challenging task to elucidate the exact genetic mechanisms underlying SCZ. Considering that most of the identified risk variants are located in the noncoding region, it is likely to be that these identified risk variants confer risk of SCZ through modulating gene expression. As the GWAS results alone could not predict whether the identified SCZ risk variants have functional consequences, a statistical approach is needed to integrate information from all disease associations and independent expression QTL data. In this study, we used *Sherlock* to systematically integrate SNP association signals from the largest GWAS of SCZ (PGC2)[20] to date and independent brain expression QTL data. We identified 10 risk genes whose mRNA expression change may contribute to risk of SCZ. To further pinpoint the most promising candidates, we performed

multiple prioritizations and comprehensive integrative analyses. Our results revealed that *ALMS1*, *CSNK2B*, and *GLT8D1* genes may represent promising risk genes for SCZ. First, *Sherlock* integrative analysis[25] showed these three genes are significantly associated with SCZ. Second, network-based prioritization (PPI and co-expression) suggested that *ALMS1* and *CSNK2B* are the most possible candidates among the top 10 genes identified by *Sherlock*. Third, these three genes were dysregulated in SCZ cases. Fourth, we successfully replicated *ALMS1*, *CSNK2B*, and *GLT8D1* as SCZ risk genes using independent and larger brain eQTL data set ($N = 467$) from Fromer et al.[26]. Fifth, we validated *ALMS1* and *GLT8D1* as SCZ risk genes using independent integrative analysis method (i.e., *SMR*)[29]. Sixth, *ALMS1*, *CSNK2B*, and *GLT8D1* are expressed in developing human brain, and disruption of *Alms1* and *Csnk2b* in mice lead to severe neurodevelopmental disorder[57]. Seventh, we showed that knockdown of *GLT8D1* and *CSNK2B* affects

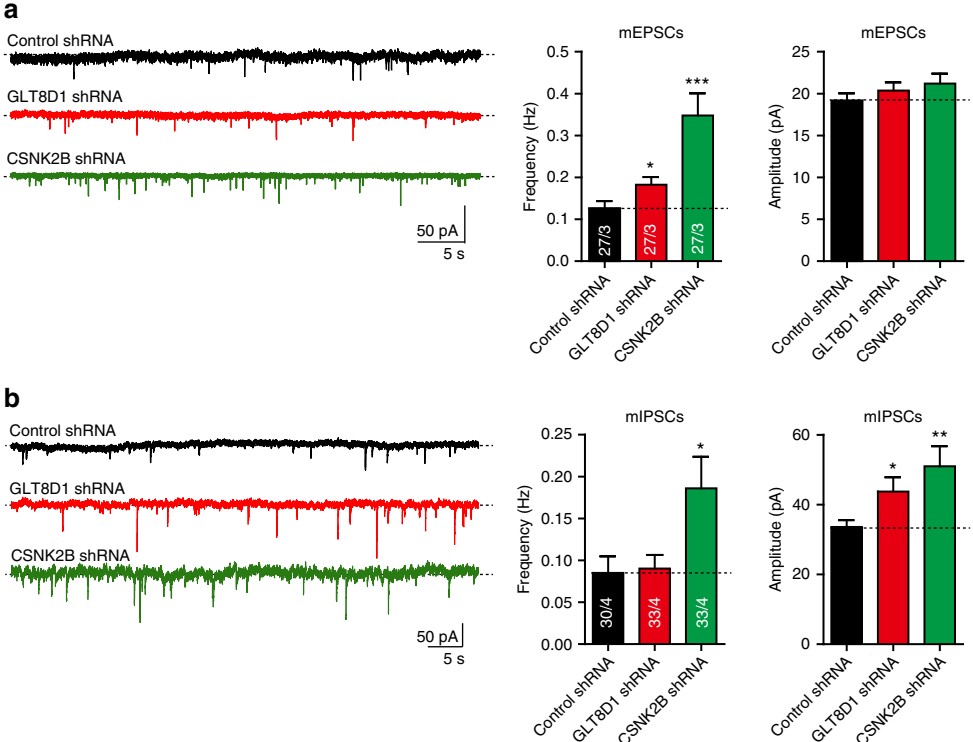

**Fig. 6** Knockdown of *GLT8D1* and *CSNK2B* alters synaptic transmission. **a** Compared with control neurons, knockdown of *GLT8D1* and *CSNK2B* affected the frequency of miniature excitatory postsynaptic currents (mEPSCs) significantly. However, the amplitude of mEPSCs was not affected. **b** Compared with controls neurons, the amplitude of miniature inhibitory postsynaptic currents (mIPSC) was significantly affected in neurons transfected with *GLT8D1* and *CSNK2B* shRNAs. The frequency of mIPSC was also significantly altered in neurons transfected with *CSNK2B* shRNAs compared with controls. A total of 27 neurons (from three independent groups) were recorded for mEPSC. For mIPSC, 30 neurons (from four independent groups) were recorded for control group, and 33 neurons (from four independent groups) were recorded for *GLT8D1* and *CSNK2B* knockdown groups. $*P < 0.05$, $**P < 0.01$, $***P < 0.001$, Student's *t*-test. Data represent mean ± SD

proliferation, self-renewal, and differentiation of cultured NSCs. Eighth, we found that knockdown of *GLT8D1* and *CSNK2B* alters morphology and synaptic transmission of neurons. Finally, CFG prioritization[56] suggested that *ALMS1*, *CSNK2B*, and *GLT8D1* are the most promising candidate genes for SCZ. These convergent lines of evidence strongly suggest that *ALMS1*, *CSNK2B*, and *GLT8D1* may have a role in SCZ pathogenesis through modulating neurodevelopment.

*CSNK2B* is located in the MHC region. Genetic variants in MHC region showed the most significant association with SCZ in a recent GWAS of SCZ (PGC2)[20]. However, due to the high level of linkage disequilibrium in MHC region, it is challenging to pinpoint the causal genes at this region. *GLT8D1* is located in chromosome 3 (p21.2) and genetic variants near *GLT8D1* showed genome-wide significant association with SCZ. Nevertheless, this locus also contains several genes, including *PBRM1*, *GNL3*, *GLT8D1*, *SNORD19*, *NEK4*, *ITIH1*, *ITIH3*, and *ITIH4*. *ALMS1* is located in Chromosome 2 (p13.1). Even though no genetic variants in *ALMS1* showed genome-wide significant association with SCZ in PGC2, SNP rs56145559 in *AMLS1* showed a trend of association ($P = 8.42 \times 10^{-8}$). Our study suggests that *ALMS1*, *GLT8D1*, and *CSNK2B* may represent promising candidate genes for these reported risk loci.

The function study of *ALMS1* and *CSNK2B* in CNS development also provides support for the potential involvement of the identified risk genes in SCZ. Heydet et al.[58] found that *Alms1* is localized at the base of cilia in hypothalamic neurons and a truncating mutation of *Alms1* reduces the number of hypothalamic neuronal cilia in mice. Intriguingly, ALMS1 is physically interacted with PCM1, a protein encoded by a SCZ risk gene[59].

These studies suggested that *ALMS1* may have a role in brain development. Recently, Huillard et al.[57] showed that *Csnk2b* has an essential role in brain development. They found that conditional knockout of *Csnk2b* in mice lead to defects in differentiation of embryonic NSCs. Moreover, they found that *Csnk2b* acts as a key regulator that positively regulates the development of oligodendrocyte precursor cells through interacting with transcription factor Olig2. These studies indicate that *ALMS1* and *CSNK2B* have essential roles in brain development. Mutation or disruption of *ALMS1* and *CSNK2B* lead to severe neurodevelopmental disorder, indicating these two genes are essential for neurodevelopment[49,57].

Even though the pathophysiology of SCZ remains elusive, the neurodevelopmental hypothesis of SCZ receives much attention[30]. Accumulating evidence suggests that SCZ is caused by abnormal neurodevelopment[60]. In fact, several recent investigations have shown that SCZ risk genes such as *DISC1* and *RELN* regulate proliferation and migration of neural progenitor cells[31,61], providing further evidence that supports the neurodevelopmental hypothesis of SCZ. If the candidates identified by *Sherlock* are authentic SCZ risk genes, they may be involved in neurodevelopment (not every risk gene follows this rule, but some of them may be). Similar to the reported SCZ risk gene *DISC1* and *RELN*, Huillard et al.[57] found that *Csnk2b* regulates proliferation and differentiation of NSCs in mice. In addition, they also showed that *Csnk2b* has a key role in oligodendrogenesis through regulating the activity of the lineage-specific transcription factor OLIG2. Intriguingly, Mauney et al.[62] found that OLIG2-immunoactive cells was significantly reduced in SCZ cases compared with controls and the differentiation of

oligodendrocyte precursors is impaired in SCZ. Consistently, Voineskos et al.[63] also found that oligodendrocyte pathway has an important role in SCZ. Of note, we also found that *CSNK2B* knockdown impaired the differentiation of NSCs into oligodendrocytes (Fig. 4c, d). These consistent and convergent lines of evidence strongly suggest that *Csnk2b* may be involved in SCZ through modulating the Olig2 and oligodendrocyte pathway.

We noticed that *GLT8D1* was also associated with bipolar disorder in a previous GWAS[64]. Considering the high genetic overlap between SCZ and bipolar disorder (among the five psychiatric disorders, including SCZ, bipolar disorder, attention-deficit/hyperactivity disorder, major depressive disorder, and autism spectrum disorders, SCZ and bipolar disorder have the highest genetic overlap)[65], the association between *GLT8D1* and bipolar disorder suggests that *GLT8D1* may also have a role in SCZ. As a matter of fact, *GLT8D1* reached a genome-wide significance level in PGC2[20], strongly suggesting that it is a true risk gene for SCZ. A recent study found that rs2535629, which showed strong association with susceptibility to psychiatric disorders in Caucasian populations[66], is also significantly associated with the expression level of *GLT8D1* gene[67]. Interestingly, rs2535629 is also significantly associated with SCZ in Japanese populations[67]. This independent evidence strongly suggests that several independent genetic variants may confer risk of SCZ through altering the expression level of *GLT8D1* gene. Consistently, we found the expression of *GLT8D1* is significantly downregulated in SCZ cases compared with healthy controls. These lines of evidence provide further support that the predicted genes may represent authentic susceptibility genes for SCZ.

It should be noted that the SNP associations used in this study were primarily from populations of European ancestry. Intriguingly, a recent large-scale GWAS study performed in Chinese population (7,699 SCZ cases and 18,327 controls) identified several novel risk loci for SCZ[19]. We examined the associations between eSNPs of *ALMS1*, *GLT8D1*, and *CSNK2B* and SCZ (Table 1) in the transancestry GWAS meta-analysis (PGC2-plus-Chinese, 43,175 cases and 65,166 controls) and found that eSNPs of *GLT8D1* and *CSNK2B* have a smaller *P*-values (*P*-value of rs6795646 is $4.56 \times 10^{-10}$ and *P*-value of rs2075800 is $1.56 \times 10^{-13}$) in the transancestry GWAS meta-analysis compared with PGC2 (SNP rs6753344 (eSNP of *ALMS1*) is not available), indicating that these two SNPs have the same risk alleles in Chinese population. These data provide further support for the involvement of *GLT8D1* and *CSNK2B* in SCZ.

One of the most interesting findings of this study is that we found that *GLT8D1* regulates proliferation, self-renewal, and differentiation of NSCs in vitro. The function of *ALMS1* and *CSNK2B* has been characterized. However, little is known about the role of *GLT8D1* in nervous system. For the first time, we showed that *GLT8D1* has active roles in nervous system through regulating proliferation, self-renewal, and differentiation of NSCs. We also showed that *GLT8D1* has pivotal roles in regulating the growth of dendrites and synaptic transmission. Moreover, this study also provides a paradigm to illuminate the genetic mechanisms underpinning complex psychiatric disorders. Despite the fact that more than 100 risk loci show robust association with SCZ[20], how the identified genetic variants contribute to SCZ risk remains largely unknown. Findings from genetic studies alone are insufficient to elucidate the detailed function and exact mechanisms of the identified risk variants and genes in SCZ. Integrating the evidence from different layers of prioritization (e.g., CFG[56]) may help to understand the genetic and pathophysiological mechanisms of SCZ, at least for pinpointing the potential risk genes of SCZ. In fact, previous studies had used

CFG to integrate evidence from different layers and they have successfully prioritized multiple promising risk genes for SCZ[56,68]. We also used CFG developed by Ayalew et al.[56] to prioritize the identified candidate genes identified in this study and found that *ALMS1*, *GLT8D1*, and *CSNK2B* have the highest score, providing further support for the involvement of these three genes in SCZ.

As SCZ is a brain disorder that mainly originates from dysfunction of brain function, we only used brain eQTL to perform *Sherlock* integrative analysis[25] in this study. Considering the relatively small sample size used in brain eQTL data set, only limited risk genes were identified by using brain eQTL. For example, we only detected 10 statistically significant risk genes using brain eQTL from Myers et al.[24] (Table 1). Recent study showed that eQTL from other tissues may also provide valuable information for identifying SCZ risk genes. Hauberg et al.[69] conducted a large-scale study recently to identify genes whose expression level were affected by risk variants through using eQTL data from multiple tissues (including brain tissues, blood, heart, skin, etc.). They integrated genetic association signals from PGC2[20] and eQTL data from multiple tissues using SMR[29] and identified numerous SCZ risk genes, which greatly expanded the number of SCZ risk genes. The study of Huaberg et al.[69] provided promising candidate genes for SCZ and more work is needed to characterize these identified risk genes in the pathogenesis of SCZ.

In summary, our comprehensive study identifies *ALMS1*, *GLT8D1*, and *CSNK2B* as SCZ risk genes whose expression level may contribute to SCZ risk. Our study links SCZ risk variants to specific genes and therefore provides a possible mechanistic explanation between genetic variation and SCZ susceptibility. This study links some of the risk variants from the largest GWAS of SCZ to specific genes, which not only provides a framework to investigate how genetic variants contribute to SCZ risk through modulating gene expression but also provides a starting point to dissect the possible role of the identified genes in the pathophysiology of SCZ.

## Methods

**Overview of the study design.** To identify genes whose expression change may confer risk of SCZ, we systematically integrated genetic association signals from the largest SCZ GWAS so far (PGC2)[20] and brain eQTL data[24] by using a Bayesian statistical framework (*Sherlock*)[25]. We identified 10 promising SCZ susceptibility genes whose expression may contribute to SCZ risk. To further prioritize the most possible candidate genes, we performed comprehensive analyses, including spatiotemporal expression pattern analysis, PPI analysis, co-expression analysis, and differential expression analysis, as described in our recent study[70].We validated our findings using independent brain eQTL data and integrative analysis method (*SMR*)[29]. Furthermore, we also showed that the identified risk genes *GLT8D1* and *CSNK2B* regulate proliferation and differentiation of NSCs, knockdown of *GLT8D1* and *CSNK2B* affects morphology and synaptic transmission of neurons. For clarity, we provided a workflow diagram to illustrate these analyses in Supplementary Fig. 1.

**SCZ GWAS data.** Recently, the PGC reported the largest SCZ GWAS so far (PGC2 release)[20]. In brief, genome-wide genotypes from 49 independent samples (46 of European and 3 of Asian ancestry, with a final sample size up to 35,476 SCZ cases and 46,839 controls) were combined and systematically meta-analyzed. The study identified 128 independent associations that reached genome-wide significant level and most of the significant associations (83 of 128) are newly reported in that study. Summary statistics (genome-wide SNP associations, i.e., *P*-values) from PGC2[20] were used as input in this study. More detailed information about sample description, diagnosis, genotyping, and statistical analyses can be found in the original study and PGC website (http://www.med.unc.edu/pgc/)[20].

**Brain eQTL data.** Brain eQTL from Myers et al.[24] were used as discovery eQTL. Briefly, Myers et al. collected 193 neuropathologically normal (i.e., without clinical history of psychiatric disorders and other neurological diseases) human brain samples (cortex) and obtained transcriptome data (using the Illumina microarrays) and genome-wide genotype data (using the Affymetrix SNP Array). Allelic test of association was conducted to test the association between each transcript and SNP

through using the PLINK[71]. More detailed information on sample collection, demographic description, gene expression quantification, SNP genotyping, and statistical analyses can be found in the original study of Myers et al.[24]. We also used brain eQTL from Common Mind Consortium (CMC)[26] and GTEx[28] to validate our results. For GTEx eQTL data (v6)[28], brain tissues (the frontal cortex) of 92 individuals were used and gene expression was quantified using RNA-sequencing technology. SNP genotyping was performed using the Illumina OMNI 5 M SNP Array. In addition, eQTL analysis was performed by Matrix eQTL[72] (assuming an additive model) using linear regression analysis, with covariates included in the analysis. For CMC eQTL data[26], brain tissues (dorsal lateral prefrontal cortex) of 258 SCZ cases and 279 controls (all of European ancestry) were used and gene expression was quantified using RNA sequencing approach and genotyping was performed using the Illumina Infinium HumanOmniExpressExome 8 v 1.1b chip. After quality control, gene expression data and genotypes of 467 subjects were used to generate brain eQTL resource.

**Sherlock integrative analysis**. On the basis of the underlying assumption that the expression change of a specific gene may contribute to SCZ risk, we used *Sherlock* integrative analysis method (http://sherlock.ucsf.edu/) developed by He et al.[25] to integrate SNP associations from PGC2[20] and brain eQTL from Myer et al.[24]. The *Sherlock* statistical inference procedures are as follows: *Sherlock* first identifies expression-associated SNPs (eSNPs) in the human brain by using the brain eQTL data from Myers et al.[24]. *Sherlock* then evaluates the association between eSNPs and SCZ using the genome-wide SNP associations from PGC2[20]. *Sherlock* scores each eSNP according to the association significance between the eSNP and SCZ. If an eSNP is significantly associated with SCZ in PGC2[20], a positive score would be assigned. If an eSNP is not associated with SCZ, a negative score would be given. If an SNP showed a significant association with SCZ only and was not associated with gene expression, this SNP will not be used in scoring. For each gene, *Sherlock* performs a Bayesian inference to test whether the expression change of this gene has any impact on the risk of SCZ by using the collective information of the putative eSNPs of the gene. Based on the combined evidence from GWAS and brain eQTL, *Sherlock* infers SCZ-associated genes through calculating the logarithm of Bayes factor of each gene. Compared with traditional analysis, which usually ignores SNPs with moderate association (e.g., SNPs with *P*-values ranging from $1 \times 10^{-6}$ to $1 \times 10^{-8}$), *Sherlock* utilizes both strong and moderate SNPs in the eQTL and GWAS data through using a powerful statistical model. *Sherlock* makes statistical inference by aggregating the information from both strong SNPs and moderate SNPs (strong SNPs have larger contribution to the final score). Bonferroni correction was used to correct the *P*-value of genes identified by *Sherlock* integrative analysis. The corrected threshold of *P*-value is $0.05/8,998 = 5.6 \times 10^{-6}$ (there were 8,998 genes in the eQTL test). More detailed information about *Sherlock* statistical framework and inference can be found in the original paper[25].

**Sherlock analysis using independent brain eQTL data**. To further validate whether the prioritized risk genes are authentic SCZ susceptibility genes, we carried out *Sherlock* integrative using an independent and much larger brain eQTL data (CMC, a total of 467 subjects were included) from Fromer et al.[26]. In addition, we also used brain eQTL from the GTEx[28] to verify *Sherlock* results.

**Integration of SCZ GWAS and brain eQTL using SMR**. We used independent integrative analysis (*SMR*) approach developed by Zhu et al.[29] to validate our results. Similar to *Sherlock*, *SMR* performs integrative analysis through integrating genetic associations from GWAS and eQTL data. In brief, if a genetic variant was an eQTL of a specific gene (i.e., this genetic variant is associated with the expression level of this gene), the individuals with different genotypes of this genetic variant will have different expression level of this gene. If the expression alteration of this gene influences disease susceptibility, then we will observe phenotype differences among subjects carrying different genotypes of this genetic variant. That is, this genetic variant is also associated with disease. Of note, *SMR* uses a different algorithm and strategies to make the statistical inference. More detailed information about statistical inference of *SMR* can be found in the original paper[29].

**Spatio-temporal expression pattern analysis of risk genes**. To explore whether the SCZ candidate genes identified by *Sherlock* are expressed in human brain, we assessed the expression profiling of these genes in diverse human brain tissues using the Gene Enrichment Profiler[33]. We also explored the spatio-temporal expression pattern of these genes in developing an adult human brain using expression data (based on RNA sequencing) from the Allen Institute for Brain Science (BrainSpan: Atlas of the Developing Human Brain (http://www.brainspan.org/)[73]. As had been described previously[34], we divided the prefrontal cortex into four sub-regions, including DFC, MFC, OFC and VFC. Gene expression values (based on RNA sequencing) of the interested genes in developing prefrontal cortex were downloaded from the BrainSpan (http://www.brainspan.org/) (access date: 10/16/2016) (*n* = 42 individuals). The original expression values were linearly transformed using min–max standardization method with following function: $x = (x – min)/(max – min)$ (*x* represents the original expression value). The transformed expression level of each gene was ranges from 0 to 1 and ggplot2 package

(http://ggplot2.org/) in R (v3.2.5) was used to plot the spatial-temporal expression patterns of the identified risk genes.

**Significance evaluation of the observed expression pattern**. *Sherlock* integrative analysis (using brain eQTL from Myers et al.[24]) identified 10 SCZ risk genes. We analyzed the expression of these genes in developing and adult human brain and found that SCZ risk genes identified by *Sherlock*[25] have higher expression level at early developmental stage than later stage (Fig. 1). To evaluate whether the observed expression pattern is expected by chance or not (i.e., if the genes identified by *Sherlock* analysis showed higher expression at early developmental stage than later stage), we performed permutation test. Briefly, genes that match the number of *Sherlock* genes in human genome were randomly selected (8 genes each time, although *Sherlock* analysis identified 10 genes, 2 of them (*LOC376138* and *LOC375768*) were not found in the BrainSpan data set (http://www.brainspan.org/)) and their expression pattern in developing an adult human brain were assessed. A 1,000 permutations test was performed to obtain the *P*-value (the proportion of tests that have same or extreme expression pattern than the observed expression pattern).

**PPI analysis**. We explored the PPI between the proteins encoded by genes identified by *Sherlock* and SCZ risk genes identified by GWAS. SCZ significant risk genes were mainly extracted from the study of PGC2[20]. In addition, we also extracted genes identified by other GWAS from the SZDB database[70] (http://www.szdb.org/). A total of 374 genes were extracted (Supplementary Table 2).

Based on the "guilt by association" principle[74] (i.e., genes associated with a specific phenotype or disease are more likely to interact (locate closer) with each other than the random chance), we explored the PPI between the proteins encoded by genes identified by *Sherlock*[25] and SCZ risk genes identified by GWAS[20,70] using Human PPI data (accessed date: 21 October 2015) from GeneMANIA[37]. We utilized the DAPPLE program (developed by Rossin et al.[38]) to prioritize the risk genes identified by *Sherlock* analysis. To assess whether proteins encoded by genes identified by *Sherlock* are physically significantly interacted with proteins encoded by risk genes identified by GWAS of SCZ, we performed the permutation test. In brief, genes that match the *Sherlock* genes in number (i.e., eight genes, as two genes (*LOC376138* and *LOC375768*) were not found in GeneMANIA database; Table 1) were randomly selected from human genome and their physical interactions with SCZ risk genes identified by GWAS (374 genes) (Supplementary Table 2) were extracted from GeneMANIA PPI database. The total number of PPIs (i.e., connectivity) among randomly selected genes and SCZ risk genes identified by GWAS were calculated and compared with the observed values. We performed 1,000 permutations to obtain the *P*-value of the permutation test.

**Co-expression analysis**. To test whether the SCZ candidate genes identified by *Sherlock* integrative analysis[25] are significantly co-expressed with SCZ risk genes identified by GWAS, we performed co-expression analysis through using expression data from the Allen Institute for Brain Science (BrainSpan: Atlas of the Developing Human Brain (http://www.brainspan.org/static/home). To evaluate the significance of the observed co-expression (i.e., if the observed co-expression pattern is due to chance effect or not), e.g., 7 out of 10 genes identified by *Sherlock* analysis[25] were significantly co-expressed (|Pearson's correlation coefficient| > 0.8)[34] with 193 SCZ risk genes identified by GWAS) (Supplementary Fig. 5), we conducted permutation test. Briefly, genes that match the *Sherlock* genes in number (i.e., 10 genes; Table 1) were randomly selected from human genome and their co-expression with SCZ risk genes identified by GWAS (374 genes) (Supplementary Table 2) were assessed. The co-expression strength (i.e., Pearson's correlation coefficient) between each randomly selected gene and SCZ risk genes identified by GWAS were calculated. If a gene showed co-expression with a SCZ risk gene identified by GWAS (i.e., the absolute value of Pearson's correlation coefficient between this gene and the SCZ risk gene identified by GWAS is > 0.8)[34], the edge of the co-expression network (i.e., the number of connectivity) will increase by one. The total number of connectivity among randomly selected genes and SCZ risk genes identified by GWAS were calculated and compared with the observed values. A 1,000 permutations test was performed and the number of simulations with total edges (i.e., connectivity) equal to or larger than observed value was used to obtain the *P*-value of permutation test.

**Network-based prioritization**. Previously studies have shown that genes associated with a particular disease or phenotype tend to physically interact[75,76] and encode a densely interconnected molecular network. We prioritized the genes identified by *Sherlock* analysis using "guilt by association" principle[74] (i.e., disease-associated genes tend to interact (locate closer) with each other than random proteins in the PPI network). DAPPLE[38] was used to prioritize the risk genes identified by *Sherlock* analysis. DAPPLE uses a within-degree node-label permutation test to correct the prioritized genes. Briefly, DAPPLE first generates an original network using the input genes. Then a random network is built and this random network has the same number of edges, size, and per-protein binding degree as the original network. The gene names are then randomly re-assigned to nodes and 50,000 permutations (random networks) were generated. More detailed information about the permutation test can be found in the paper of Rossin et al.[38].

Similar to prioritization the identified genes using PPI data, recent studies showed that genes associated with psychiatric disorders (such as SCZ and autism) are significantly co-expressed in human brain[34,39,77]. Thus, co-expression data can also be used to prioritize SCZ risk genes. If a gene is frequently interacted or co-expressed with other SCZ genes, this gene may also be associated with SCZ (a phenomenon called "guilt by association"). Permutation test was used to test whether the prioritized risk genes are significantly co-expressed with genes identified by GWAS. More detailed information about correction of co-expression can be found in the section of co-expression analysis.

**Expression analysis in SCZ cases and controls.** *Sherlock* identifies disease-associated gene with the underlying assumption that the expression level of this gene may have a role in disease pathogenesis. To explore whether the risk genes identified by *Sherlock* integrative analysis[25] are differentially expressed in SCZ cases compared with controls, we obtained public available expression data sets, including GSE53987[40] and GSE12649[41]. GSE53987 contains brain transcriptome (including three brain regions, the prefrontal cortex, the hippocampus, and the striatum) data of 19 SCZ cases and 19 controls. GSE12649 contains brain transcriptome (the prefrontal cortex) of 35 SCZ cases and 34 controls[41]. In addition, we also examined the expression of risk genes using the Stanley neuropathology consortium integrative database[42]. We downloaded the raw expression values from GEO (https://www.ncbi.nlm.nih.gov/geo/) and performed the analysis using the parameters and statistical methods described in the original papers. More detailed information about sample collection, RNA extraction and quantification, quality control, and confounding variables can be found in the original studies[40,41].

**Association of the risk genes with hippocampal structure.** Previous studies have shown that genetic risk variants in SCZ risk genes (e.g., *BDNF* and *DISC1*) were associated with hippocampal structure in healthy subjects[44,45]. We thus explored the association between risk variants (i.e., eSNPs identified by *Sherlock* and *SMR* integrative analysis) in *ALMS1*, *GLT8D1*, and *CSNK2B*, and hippocampal structure in the ENIGMA sample (a total of 13,717 healthy subjects)[46]. To uncover the genetic variants influencing human subcortical brain structures, the ENIGMA consortium performed a large-scale meta-analysis by combing magnetic resonance imaging (MRI) scans and genome-wide genetic information from multiple sites (a total of 13,717 normal subjects)[46]. Inverse variance-weighted fixed-effect model was used to perform the meta-analysis. The associations between eSNPs of *ALMS1*, *GLT8D1*, and *CSNK2B*, and hippocampal structure were extracted from the ENIGMA website (https://www.enigma-brain.org/enigmavis/visualizer/visualizer). More detailed information about the ENIGMA (including sample description, MRI collection, genotyping, quality control, and statistical analyses can be found in the original paper[46]).

**Association of the three risk genes with cognitive function.** Recent studies have shown that SCZ risk variants are associated with cognitive function in either SCZ patients or healthy controls[78,79]. We thus investigated the association between eSNPs of *AMLS1*, *GLT8D1*, *CSNK2B*, and cognitive function in healthy subjects. We focused on working memory, a well-characterized cognitive function that has been frequently reported to be impaired in SCZ[80]. A total of 1,132 healthy Chinese college students were recruited for the cognitive function study, including 522 males and 610 females, with an average age of 21.1 ± 2.2 years. No history of psychiatric diseases, head injuries, stroke, or seizure were reported for all subjects. All participants finished two working memory tasks: verbal three-back task and spatial two-back task. In the verbal three-back task, subjects viewed a sequence of single letter presented on a computer screen one by one for 750 ms each, with intertrial interval of 2,250 ms, and judge whether the current letter was the same as the one presented three items before by pressing keys. Each sequence is composed of 13 letters, 3 letters were showed on the screen first and the participants need to remember the order of the first 3 letters, then a new letter was presented and they have to judge whether the current letter matches the one from the earlier 3 steps in the sequence. Overall, accuracy from six different sequences was used to measure the working memory performance. The spatial two-back task used a similar paradigm. Ten squares were randomly scattered on the screen (with fixed position during the whole experiment) and squares were flashed one by one in a random order for 500 ms, with intertrial interval of 1,500 ms. Participants have to memorize the position of last two flashed squares and judge whether the current flashed square is the one flashed two items before. Each sequence is composed of 12 flashes and the overall accuracy of 4 finished sequences was recorded. To make sure that the participants understand the task but not over trained, all participants practiced three sequences or got accuracy higher than 0.7 before performing each task. The Institutional Review Board (IRB) of the State Key Laboratory of Cognitive Neuroscience and Learning at Beijing Normal University reviewed and approved this protocol, and all subjects signed a consent form after a full explanation of the study procedure.

**Isolation and culture of mouse NSCs.** Mouse NSCs were isolated from brains of embryonic day 14.5 mice (from a 2-month-old female mouse, C57BL6) and grown in serum-free growth medium (Dulbecco's modified Eagle's medium (DMEM)/F12 1:1; Hyclone) containing 20 ng ml⁻¹ epidermal growth factor, 20 ng ml⁻¹ basic fibroblast growth factor, 1% penicillin/streptavidin, 1% N2 supplement (Gibco), B27 supplement (Gibco), and 10 μg ml⁻¹ heparin as previously described[81]. The resulting multipotent neurospheres were passaged every 5–7 days to single-cell suspension for continued growth and enrichment of stem cells. We also isolated NSCs from P7 dentate gyrus as previously described[82], and performed proliferation and differentiation assays. We have checked the primary cultured embryonic NSCs and there was no mycoplasma contamination.

**Knockdown of *GLT8D1* and *CSNK2B* in mouse NSCs.** shRNAs targeting to *GLT8D1* and *CSNK2B* were constructed using pLKO.1 vector and constructs were verified through sequencing. The 21 bp targeting sequences are: *GLT8D1*-shRNA#1, 5′-GATGATGATGTCATTGTACAA-3′; *GLT8D1*-shRNA#2, 5′-ACACACTATGTGGGAAGGTAA-3′; *CSNK2B*-shRNA#1, 5′-GCTCTGGACATGATCTTAGAC-3′; *CSNK2B*-shRNA#2, 5′-TTGGCTACTGTCCTCGTGTAT-3′. The control scramble shRNA sequence is: 5′-GCACTACCAGAGCTAACTCAG-3′. Lentiviruses were generated according to the manufacturer's protocol. After 72 h viral infection, cells were treated with puromycin (1 μg ml⁻¹) to select NSCs stably expressing indicated shRNAs.

**Real-time quantitative PCR.** Cells were washed once with 1 × phosphate-buffered saline (PBS), then total RNA was extracted with Trizol reagent (Takara). After RNA isolation, 1 μg total RNA was treated with gDNA Eraser (Takara) to remove the genomic DNA, followed with reverse transcription using PrimeScript RT reagent kit (Takara). We used 2 μl complementary DNA (1:5 dilution) as template and Universal SYBR Green Supermix (Roche) was used for real-time quantitative PCR (qPCR). Cycling parameters were 95 °C for 3 min, followed by 40 cycles of 95 °C for 15 s and 60 °C for 30 s. The *GAPDH* gene was used as the internal control and the delta-delta Ct method was used to quantify gene expression level. Primers used for verification the knockdown efficiency of shRNAs were: *GLT8D1*_F: 5′-GTTCCCAGTGCAAAGAAAGC-3′ and *GLT8D1*_R: 5′-TCCCCGGATGAT-GACTTTAG-3′; *CSNK2B*_F: 5′-GCAGGTGCCTCACTATCGAC-3′ and *CSNK2B*_R: 5′-CTCTGGTTGGGGTTGTCTTC-3′. Primer sequences used for other reverse transcriptase-qPCR assays are listed in Supplementary Table 8.

**Western blotting.** Whole-cell protein extracts were lysed by 1 × NP40 buffer and centrifuged at 14,000 g for 15 min. The bicinchoninic acid assay (BCA assay) was used to quantify the protein concentrations. Proteins were separated by 10% SDS-polyacrylamide gel electrophoresis, transferred to polivinylidene difluoride membrane, and blocked by 5% no-fat milk for 2 h. The membrane was incubated with primary antibodies overnight at 4 °C. The primary antibodies used in this study were as follows: Rabbit anti-GLT8D1 (1:500, Absin abs128289a), Rabbit anti-CSNK2B (1:1,000, Abcam ab76025), and mouse anti-GAPDH (1:10,000, Proteintech, 60004-1-IG). Proper secondary antibodies (1:2,000, Abclonal, AS-003, AS-014) were used at room temperature for 2 h.

**NSCs proliferation and differentiation assays.** To investigate the role of *GLT8D1* and *CNSK2B* in mouse NSCs, we conducted proliferation and differentiation assays. BrdU staining was performed as previously described[82] and NSCs were pulsed with 10 μM BrdU for 20 min to label dividing cells. The quantification of proliferation was performed manually in a blinded manner with Image J software (https://imagej.nih.gov/ij/) and Student's *t*-test. For differentiation assay, NSCs were cultured for 3 days in differentiation medium: DMEM/F12 containing 1% penicillin/streptavidin, 1% N2 supplement (Gibco), B27 supplement (Gibco), and 10 μg ml⁻¹ heparin, followed by fixation. Immunocytochemistry staining was carried out as previously described[83]. Cells were pre-blocked using PBS containing 5% normal goat serum and 0.1% Tween-20 for 30 min, incubated with indicated primary antibodies: BrdU (1:1,000, Abcam ab142567), mouse-anti BrdU (1:1,000, CST, 5292S), mouse Tuj1 (1:1,000, Sigma, T8578), rabbit GFAP (1:1,000, DAKO, Z0334), mouse O4 (1 : 1,000, R&D System, MAB1326), and MAP2 (1:1,000, Millipore, AB5622), and then incubated with secondary antibodies that included goat anti-mouse Cy3 (Abclonal, AS008) or goat anti-rabbit Cy3 (1:500, Abclonal, AS007), followed by staining with the fluorescent nuclear dye 4′,6-dimidino-2′-phenylindole dihydrochloride (Sigma, D3571). The numbers of Tuj1+, GFAP+, O4+, and MAP2+ cells were quantified with Image J software and statistical analysis was performed using two-tailed Student's *t*-test. Image-pro plus 6.0 was used for quantitative measurements of dendritic length, number of primary dendrites, and secondary dendrites. Quantification of dendritic length was performed as previously described[84] and a total of 60 neurons were measured for the control and experimental groups, respectively. All experiments were performed for three independent assays with at least three replicates per group. Dendrites-related data (including dendrites number, dendrites length, and number of branch points) were represented as mean ± SEM. Other data were represented as mean ± SD. Difference was considered to be statistically significant if the *P*-value is < 0.05.

**Electrophysiological recordings.** Primary hippocampal neurons were derived from mice pubs (0–24 h) as previously described[85,86]. Calcium phosphate method was used for the transfection of neurons at DIV (days in vitro) 10[86]. Briefly, for

each coverslip (8 mm diameter) in a 48-well plate, 0.5 μg plasmid (green fluorescent protein-tagged-pLKO.1 shRNA) was mixed with 0.992 μl 2 M $CaCl_2$ and the pre-mixed DNA/$CaCl_2$ solution was added to 8 μl of 2xHBS. The DNA/$CaCl_2$/HBS mixed solution was incubated at room temperature for 30 min, and then added to the cultured hippocampal neurons and incubated for another 30 min in the incubator, followed by further incubation for 30 min in the incubator (5% $CO_2$, 37 °C). After incubation, neurons were washed with neuronal culture medium containing 10 mM $MgCl_2$ for 15 min and cultured for 4 days before electrophysiology recordings.

Whole-cell patch clamp recordings were performed as previously described[86,87]. The pipettes were filled with solution containing (mM): 145 KCl, 1 $MgCl_2$, 5 NaCl, 5 EGTA, 0.3 $Na_2$GTP, 4 MgATP, 5 QX-314, and 10 HEPES. Neurons were maintained in solution containing (mM): 4 KCl, 150 NaCl, 2 $CaCl_2$, 1 $MgCl_2$, 10 HEPES, and 10 glucose. Multiclamp 700B and pCLAMP 10.0 (Molecular Devices, Sunnyvale, CA, USA) were used for amplifying and acquisition of the signals. Data were analyzed using Clampfit 9.02 (Molecular Devices), Igor 4.0 (WaveMetrics, Portland, USA), and GraphPad Prism 5 (GraphPad Software, La Jolla, CA, USA).

**Prioritization of the SCZ risk genes using the CFG approach**. We conducted different analyses (including *Sherlock* integrative analysis, PPI, co-expression, differential expression analysis, association with brain structure and cognitive functions, validation of *Sherlock* results using independent brain eQTL and integrative method (*SMR*), and functional assays) to identify the potential SCZ risk genes. To combine the lines of evidence from different analyses, we used the CFG[56] method to prioritize the identified genes. Briefly, CFG uses multiple independent lines of evidence to identify and prioritize SCZ risk genes. Evidence used for scoring are as follows: (1) *Sherlock* integrative analysis[25] results (brain eQTL from Myers et al.[24]); (2) PPI results; (3) co-expression results; (4) differential gene expression results; (5) *Sherlock* integrative analysis results using independent brain eQTL data (from Fromer et al.[26]); (6) *SMR* integrative analysis[29] results; (7) results from the association with brain structure; (8) results from the association with cognitive function; and (9) evidence from mouse NSCs assays. Evidence from each analysis contributes one point to the studied genes in the CFG analysis and a cumulative scoring strategy was used to calculate the final score of the studied genes (i.e., polyevidence scoring)[56]. A higher total score suggests that more evidence from different analyses supports this gene as a SCZ risk gene. More detailed information about CFG and polyevidence scoring strategy can be found in previous study[56] and Supplementary Fig. 1.

**Data availability**. Genome-wide SNP associations were obtained from PGC2 (http://www.med.unc.edu/pgc/)[20]. Brain eQTLs were from the study of Myers et al.[33] (http://labs.med.miami.edu/myers/LFuN/data.html) and GTEX (https://gtexportal.org/)[28]. Spatio-temporal expression data were from the Allen Institute for Brain Science (BrainSpan: Atlas of the Developing Human Brain (http://www.brainspan.org/). Human PPI data were from GeneMANIA (http://genemania.org/)[37]. Associaitons between the studied SNPs and hippocampal volume were from ENIGMA2 (http://www.enigma-viewer.org/About_the_projects.html)[46]. Gene expression data (GSE53987[40] and GSE12649[41]) were Gene Expression Omnibus (GEO) database (https://www.ncbi.nlm.nih.gov/geo/). *Sherlock* software: http://sherlock.ucsf.edu/25. SMR software: http://cnsgenomics.com/software/smr/29. DAPPLE software: http://archive.broadinstitute.org/mpg/dapple/dapple.php38. ImagJ: https://imagej.nih.gov/ij/. SZDB database: http://www.szdb.org/70. The results of Sherlock integrative analysis will be made available at SZDB database: http://www.szdb.org/70. The rest of the data are available from the corresponding author upon request.

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

## Acknowledgements

This study was equally supported by the National Key Research and Development Program of China (Stem Cell and Translational Research) (2016YFA0100900) and Strategic Priority Research Program of the Chinese Academy of Sciences (XDB13000000) to X.-J.L. and Y.C. This study was also was supported by the National Basic Research Program of China (2017YFA0105201), the National Nature Science Foundation of China (31722029 to X.-J.L., U1502224 and 81772996 to Y.C., 81672764 to C.-P.Y., 31670842 to C.Z., and 31571150 to C.C.), Yunnan Applied Basic Research Projects (2014FA038, 2016FA009, and 2014FB182), Yunnan province High-Level talents Introduced program 2013HA021, and the Beijing Municipal Science and Technology Commission (Z161100002616021 and Z161100000216154). C.-P.Y. was also supported by the Chinese Academy of Sciences Western Light Program, Youth Innovation Promotion Association, CAS. X.-J.L. was supported by the 1000 Young Talents Program. Y.-G.Y. was supported by the Strategic Priority Research Program (B) of CAS (XDB02020003) and the Bureau of Frontier Sciences and Education of CAS (QYZDJ-SSW-SMC005).

## Author contributions

X.-J.L. conceived and designed the study. X.-Y.L., Y.W., J.L., and Y.H. performed the integrative genomic and bioinformatic analysis. C.-P.Y., Q.S., and Y.C. conducted the proliferation, differentiation, and morphology assays of mouse NSCs, qPCR, and western blotting. M.W. and C.Z. performed the electrophysiological recordings. C.C. and G.X. performed the cognitive experiments. C.-P.Y., Y.W., Y.Z., Q.X., M.W., K.L., Y.-G.Y., C.Z., M.L., Y.C., and X.-J.L. contributed with data analysis and interpretation of the results.

X.-J.L. and Y.C. oversaw the project, designed experiments, analyzed data, and drafted the article. All authors reviewed and approved the final version of the manuscript.

## Additional information

Competing interestThe authors declare no competing financial interests.

