## [Peer Review File · Nature Communications]

Reviewer #1 (Remarks to the Author):

Yang et al. performed a mostly functional analysis integrating expression and association data for schizophrenia. This work is indeed comprehensive but still should address several points (described below) before being considered for publication.

1. lines 60-61: "Persons with schizophrenia have a high risk for suicide, which is one of the leading causes of death for people with schizophrenia." Please delete the last part of the sentence.
2. From lines 89 to 104: This section is more suited for the methods section and there is no clear description of the aim of the paper. Please clarify.
3. The Sherlock analysis is not completely described since the correction for multiple testing is not described.
4. The Spatio-temporal expression pattern analysis should be a bit better described.
5. Line 152: PPI analysis, What is PPI?
6. Line 170: Network-based prioritization. How is this analysis corrected for?
7. I see that there are a lot of non-independent analyses performed and then used as validation for the other ones. This does not seem to be completely valid. Could the authors find the same results if they only applied non-related (but complementary) analysis approaches?
8. line 284: "... analysis, a powerful statistical method to detect..." there is no power analysis supplied.
9. line 303: "...We found that most of the 10 top candidate genes identified by Sherlock..." What is most? Also, gene expression at the brain level is not a good filtering approach since a significant number of genes across the genome express in brain and DOES NOT imply brain development. Please correct

10. The PPI results are not significant hence, not supporting that the genes identified by Sherlock may have a role in schizophrenia. Please correct this.

11. Are the brain tissue expression comparisons adjusted for multiple testing (multiple genes in multiple tissues? Also, sample size of this group is quite small (19 per group or 34 per group?) what a priori power is there to find these effects?

12. where the same brain tissues available for both brain tissue sets? (GSE53987 and GSE12649) If yes, was there any replication of findings?

13. The results presented on line 387 should be presented earlier (as replication of initial results) please revise

14. It is not clear why the CSNK2B gene is followed further if the expression results are negative for this gene.

15. The CFG analysis appears a bit misleading since it is not clear what information on which genes is being used. If the authors only used the info on the three best candidates described throughout the paper, then it is obvious that these should get the best results (circular analysis). If other genes were included, where are the results of the other analysis?

15. The discussion is a bit confusing. It is not clear to me the rationale to perform all these analyses. It is clear that many of these steps are not independent from each other and they should be combined.

16. What is the variance explained of SCZ, or any of the brain traits tested by the 3 genes discovered by the authors?

The paper will benefit tremendously from a figure (of sorts) walking the reader through all the steps done.

Reviewer #2 (Remarks to the Author):

The manuscript by Luo and colleagues performed comprehensive analysis of genetic and genomic data related to SCZ, and discovered three new risk genes. The authors started with integrating brain eQTL and SCZ GWAS data using a statistical tool, Sherlock, and identified 10 candidate genes. With additional analysis, they focused on three of these genes, and supported their relevance in SCZ with a number of lines of evidence, such as temporal expression patterns in brain and association with related psychiatric traits. Overall, the results supporting these genes are quite overwhelming. Most interestingly, GLT8D1 seems a very novel gene whose function was largely unknown before this study. The authors provided clear experiment evidence of this gene using Neural Stem Cell model.

Comments:

The novelty of the findings is not entirely clear. Two of these genes, CSNK2B and GLT8D1, seem to have high-confidence GWAS loci nearby (Table 1). In the published GWAS paper of SCZ, what were the genes assigned to these two loci? Or is there any published study that has already linked these two genes with the two GWAS loci? A related question is that the authors used a list of GWAS significant genes. Is this list based on the nearest genes of GWAS loci?

The authors made statements in quite a few places without statistical justifications:

303: "most of the 10 top candidate genes identified by Sherlock were widely expressed in human brain regions": Should quantify the statistical evidence here.

323: "frequently interacted with proteins encoded by GWAS significant schizophrenia risk genes". No statistical evidence/p-value.

310: "Permutation test showed that the observed expression pattern is unlikely due to chance effect". What is being tested here? The details of this test should be provided in the text (at least in Methods), not in supplement.

161: Co-expression analysis. It is not clear what the authors tested: number of genes co-expressed with known SCZ genes? Or the average level of co-expression? The key details should be provided here.

Reviewer #3 (Remarks to the Author):

In this study, Yang et al. integrated association signals of a largest GWAS report (PGC2) and brain eQTL data, and carried out a series of statistical analysis. In total, 10 candidate causal genes were predicted, while 3 of them, ALMS1, CSNK2B and GLT8D1 were highlighted by the authors, with validation analyses with different methods or in independent data. The current manuscript was well written, and the analysis is sound. I only have the following 2 points:

1. The brain eQTL data mainly used in this study included 193 normal human subjects, which is relatively smaller than a recent publication (Mads et al. Am J Hum Genet. 2017 Jun 1;100(6):885-894.), which integrated GWASs with 24 studies of expression quantitative trait loci. In addition, PGC2 schizophrenia dataset is also one of the studied 57 GWASs in the Mads et al's report. It will be better for the authors to refer to these studies and discuss the advantages and shortcomes of the current study.
2. I am wondering whether the authors have western blotting results after transfections the RNAi vectors in the cell-based assay, which might provide clearer evidences for the related downstream pathways after knocking-down GLT8D1.

Reviewer #4 (Remarks to the Author):

Peng et al performed GWAS studies and identified 10 candidate genes that contribute to schizophrenia risk. Of these 10 candidate genes, the authors identified ALMS1, CSNK2B and GLT8D1 as the most promising candidates and validated their expression in the human schizophrenia patient cases. Furthermore, they demonstrated that GLT8D1 knockdown using shRNA promotes the self-renew and proliferation abilities of cultured neural stem cells (NSCs) and inhibits NSCs differentiation into astrocytes and neurons, providing the evidence of the potential role of identified risk genes in neurodevelopment. Overall, this study identified novel schizophrenia risk genes from GWAS analysis, validated their expression and hippocampal structures in human schizophrenia patient cases. However, I have major concerns of the functional studies they performed in this study. The points are listed as below:

1. The authors performed all of their analysis in human cases, including GWAS, candidate gene expression analysis, and hippocampal structure analysis but the functional studies were performed in the mouse early postnatal (P7) NSC cultures. It is unclear whether human NSCs express these genes and whether the expression pattern of these genes in the mouse hippocampus is similar to that in human. Moreover, it is unclear why postnatal mouse NSC culture was used, since schizophrenia was considered as a neurodevelopmental disorder. Mouse embryonic culture appears to be more relevant. To best validate the neurodevelopmental deficits associated with human schizophrenia, patient-derived induced pluripotent stem cell (iPSC) culture should be considered in this context.

2. The authors state that ALMS1, CSNK2B and GLT8D1 play active roles in brain development, and showed that reduced GLT8D1 expression led to deficits in NSC proliferation and differentiation. However, they have not shown whether ALMS1 and CSNK2B exhibit similar NSC phenotypes. Despite that other studies showed the role of ALMS1 and CSNK2B in the central nervous systems, these phenotypes are not relevant to hippocampal structure deficits observed in schizophrenia patient cases.

3. The authors showed GLT8D1 knockdown in mouse NSCs led to increased self-renewal, and decreased neuronal and glial differentiation. The authors used GFAP marker for astrocytes. However, in the NSC culture, it has been widely accepted that NSCs give rise to oligodendrocytes besides neurons and astrocytes. This oligo marker should be quantified as well.

4. Schizophrenia is considered as a synaptic disorder characterized by aberrant synapses (both excitatory and inhibitory). Dendritic and synaptic phenotypes should be included in the analysis associated with GLT8D1 knockdown, including dendritic length, branches, crossings, and functional synaptic characterizations (mIPSCs and mEPSCs).

Response to referees

We thank the reviewers greatly for their thorough reviews and highly appreciate the comments and suggestions, which significantly contributed to improving the quality of this manuscript. Please find below a detailed point to point response to each comment. The revised information shown in Page and reciprocal Figures was marked in red below.

Reviewer #1 (Remarks to the Author):

Yang et al. performed a mostly functional analysis integrating expression and association data for schizophrenia. This work is indeed comprehensive but still should address several points (described below) before being considered for publication.

1. lines 60-61: "Persons with schizophrenia have a high risk for suicide, which is one of the leading causes of death for people with schizophrenia." Please delete the last part of the sentence.

Response: We thank the reviewer for pointing this out. Following the reviewer's suggestion, we have deleted the last part of the sentence. Page 5, line 7.

2. From lines 89 to 104: This section is more suited for the methods section and there is no clear description of the aim of the paper. Please clarify.

Response: We thank the reviewer for this valuable suggestion. Following the reviewer's suggestion, we have moved this section to methods. Page 7, lines 5-18.

In addition, we also added a paragraph to describe the aim of the paper. Page 6, lines 13-21, Page 7, lines 1-2.

3. The sherlock analysis is not completely described since the correction for multiple testing is not described.

Response: We thank the reviewer for pointing this out. According to the reviewer's suggestion, we have described the correction for multiple testing in detail in the revised manuscript. Page 9, lines 11-13.

4. The Spatio-temporal expression pattern analysis should be a bit better described.

Response: We appreciate the reviewer for this suggestion. Following the reviewer's suggestion, we provided the detailed information about spatio-temporal expression pattern analysis in the revised manuscript. Page 10, lines 12-21, Page 11, lines 1-2.

5. Line 152: PPI analysis, What is PPI?

Response: We apologize for the confusion, PPI is abbreviation of protein-protein interaction, and we have clarified this in the revised manuscript. **Page 11, line 18.**

6. Line 170: Network-based prioritization. How is this analysis corrected for?

Response: We thank the reviewer for pointing this out. Following the reviewer's suggestion, we provided more detailed information about the correction of network-based prioritization in the revised manuscript. **Page 13, lines 19-22, Page 14, lines 1-3, Page 14, lines 8-11.**

7. I see that there are a lot of non-independent analysis performed and then used as validation for the other ones. This does not seem to be completely valid. Could the authors find the same results if they only applied non-related (but complementary) analysis approaches ?

Response: We appreciate the reviewer for this valuable suggestion. According to the reviewer's suggestion, we only included the non-related (but complementary) analysis approaches and performed the integrative analysis again. The detailed procedures are as follows: First, we performed *Sherlock* analysis twice by using independent brain eQTL datasets, and only retained the result from the initial stage for subsequent prioritization. Second, we used an independent integrative approach (i.e., *SMR*) and performed integrative analysis on the same eQTL dataset, following the reviewer's suggestion, we also excluded the result from *SMR* and performed prioritization. Therefore, results from step 5 and step 6 in Supplementary Figure S1 were excluded from prioritization. Consistent with our original findings, we obtained the same results (i.e., *ALMS1*, *CSNK2B* and *GLT8DI* are the risk genes with highest prioritization scores). More detailed information about prioritization can be found in Supplementary Material. **Supplementary Figure S1.**

8. line 284: "... analysis,³⁴ a powerful statistical method to detect..." there is no power analysis supplied.

Response: We thank the reviewer for pointing this out. We checked the original *Sherlock* paper carefully and found that there is no power analysis supplied in the original *Sherlock* paper. We thus revised this accordingly in the revised manuscript. **Page 20, lines 13-15.**

9. line 303: "...We found that most of the 10 top candidate genes identified by Sherlock..." What is most ?

Response: Following the reviewer's suggestion, we have added the exact number of genes in the revised manuscript. **Page 22, line 22.**

Also, gene expression at the brain level is not a good filtering approach since a significant number of genes across the genome express in brain and DOES NOT imply brain development. Please correct

We thank the reviewer for pointing this out. Following the reviewer's suggestion, have corrected this in the revised manuscript. **Page 22, line 22, Page 23, lines 1-2.**

10. *The PPI results are not significant hence, not supporting that the genes identified by Sherlock may have a role in schizophrenia. Please correct this.*

Response: We thank the reviewer for pointing this out. According to the reviewer's suggestion, we have corrected this in the revised manuscript. **Page 24, lines 3-6.**

11. *Are the brain tissue expression comparisons adjusted for multiple testing (multiple genes in multiple tissues ? Also, sample size of this group is quite small (19 per group or 34 per group?) what a priori power is there to find these effects ?*

Response: We understand the reviewer's concern. Brain tissue expression comparisons were not adjusted for multiple testing. However, convergent lines of evidence suggest dysregulation of *ALMS1* gene in schizophrenia cases. First, *ALMS1* was significantly up-regulated in hippocampus of schizophrenia cases ($P=0.006$, GSE53987) compared with controls, while *CSNK2B* ($P=0.005$, GSE53987) and *GLT8D1* ($P=0.006$, GSE53987) were significantly down-regulated in hippocampus of schizophrenia cases. When corrected for multiple testing (Benjamini–Hochberg procedure), these three genes still showed significant dysregulation in hippocampus of schizophrenia cases. Second, *ALMS1* is consistently up-regulated in schizophrenia cases in four independent expression datasets (**Figure 3A, Supplementary Figure S5 and S6**), suggesting the dysregulation of this gene in schizophrenia cases.

12. *where the same brain tissues available for both brain tissue sets? (GSE53987 and GSE12649) If yes, was there any replication of findings?*

Response: Both GSE53987 and GSE12649 contain the prefrontal cortex tissues from schizophrenia cases and controls. *ALMS1* showed a trend of up-regulation ($P=0.056$) in GSE53987 (**Figure 3A, middle panel**). In GSE12649, *ALMS1* was also significantly up-regulated ($P=0.019$) (**Supplementary Figure S5**). GSE53987 also includes hippocampal tissues, *CSNK2B* ($P=0.005$) and *GLT8D1* ($P=0.006$) were significantly down-regulated in hippocampus of schizophrenia cases (**Figure 3B,C**). The expression of *CSNK2B* and *GLT8D1* in the prefrontal cortex tissues did not show significant difference in GSE53987 and GSE12649.

13. *The results presented on line 387 should be presented earlier (as replication of initial results) please revise*

Response: We thank the reviewer for this valuable suggestion. Following the reviewer's suggestion, we have revised the manuscript accordingly. **Page 21, lines 1-22, Page 22, lines 1-11.**

14. It is not clear why the *CSNK2B* gene is followed further if the expression results are negative for this gene.

Response: We understand the reviewer's concern. To identify genes whose expression change may contribute to schizophrenia risk, we first used *Sherlock* to integrate genetic associations from GWAS and brain eQTL. Based on the 10 top genes identified by *Sherlock* integrative analysis, we then used the convergent functional genomics method to prioritize the most promising risk genes for schizophrenia among these 10 genes (**Supplementary Figure S1**). As stated above, expression of *CSNK2B* gene was significantly down-regulated in hippocampus of schizophrenia cases compared with controls. In addition, we used evidence from different layers (e.g., PPI, co-expression, association with brain structure and cognitive functions, etc) to prioritize the top genes. That is, evidence from each layer only contributes one point to the final score of each gene. Based on the above evidence, *CSNK2B* may represent a promising candidate risk gene for schizophrenia.

15. The CFG analysis appears a bit misleading since it is not clear what information on which genes is being used. If the authors only used the info on the three best candidates described throughout the paper, then is obvious that these should get the best results (circular analysis). If other genes were included, where are the results of the other analysis ?

Response: We are sorry for the confusion. We first identified genes whose expression change may confer schizophrenia risk using *Sherlock* integrative analysis. We identified 10 top genes (corrected $P < 0.05$) (**Table 1a**). To prioritize the most promising risk genes among the 10 top genes, we used CFG analysis to integrate the evidence from different layers. In fact, we included the top 10 genes through the analysis and the score of each gene was calculated at each prioritization step. For clarity, we included a flowchart to illustrate the detailed prioritization procedure (**Supplementary Figure S1**).

16. The discussion is a bit confusing. Is not clear to me the rationale to perform all these analysis. It is clear that many of these steps are not independent from each other and they should be combined.

Response: We are sorry for the confusion. According to the reviewer's comment, we only included the non-related (but complementary) analysis approaches and performed the integrative analysis again. The detailed procedures are as follows: First, as we performed *Sherlock* analysis twice by using independent brain eQTL datasets, we only retained the result from the initial stage for subsequent prioritization. Second, we used an independent integrative approach (i.e., *SMR*) and performed integrative analysis on the same eQTL dataset, following the reviewer's suggestion, we also excluded the result from *SMR* and performed prioritization. That is, results from step 2 and step 3 in Supplementary Figure S1 were excluded from prioritization. Consistent with our original findings, we obtained same results (i.e., *ALMS1*, *CSNK2B* and *GLT8D1* are the risk genes with highest prioritization scores). More detailed information about prioritization can be found in Supplementary Material. **Supplementary Figure S1**.

17. What is the variance explained of SCZ, or any of the brain traits tested by the 3 genes discovered by the authors?

Response: We are sorry that we could not calculate the variance explained by the three genes as we could not access the genotype data of the significant SNPs near these three genes. However, in the original paper by PGC2 (1), the authors performed polygenic risk score profiling and they found that about 3.4% SCZ variance can be explained by genome-wide significant loci (assuming independent SNP effects). Considering that 108 independent risk loci have been identified by PGC2, each locus only explains very small variance ($3.4\%/108=0.031\%$). Assuming independent SNP effects, the variance explained by these three risk loci is less than 0.094%. Nevertheless, it should be noted this is an estimate, as we could not access the original genotype data, we could not do an accurate calculation.

The paper will benefit tremendously from a figure (of sorts) walking the reader through all the steps done.

Response: We thank the reviewer for this valuable suggestion. Following the reviewer's suggestion, we included a flowchart to illustrate the workflow of this study. **Supplementary Figure S1.**

Reviewer #2 (Remarks to the Author):

The manuscript by Luo and colleagues performed comprehensive analysis of genetic and genomic data related to SCZ, and discovered three new risk genes. The authors started with integrating brain eQTL and SCZ GWAS data using a statistical tool, Sherlock, and identified 10 candidate genes. With additional analysis, they focused on three of these genes, and supported their relevance in SCZ with a number of lines of evidence, such as temporal expression patterns in brain and association with related psychiatric traits. Overall, the results supporting these genes are quite overwhelming. Most interestingly, GLT8D1 seems a very novel gene whose function was largely unknown before this study. The authors provided clear experiment evidence of this gene using Neural Stem Cell model.

Response: We thank the reviewer for the positive comment on our work.

Comments:

The novelty of the findings is not entirely clear. Two of these genes, CSNK2B and GLT8D1, seem to have high-confidence GWAS loci nearby (Table 1). In the published GWAS paper of SCZ, what were the genes assigned to these two loci?

Response: We understand the reviewer's concern. *CSNK2B* is located in MHC region (Chr 6). Genetic variants in MHC region showed the most significant association with schizophrenia in a recent GWAS of schizophrenia (PGC2) (1). However, due to the high

level of linkage disequilibrium in MHC region, it is challenging to pinpoint the causal genes at this locus. A recent study showed that *C4A* (located in MHC region) may represent an authentic causal gene for schizophrenia at MHC region (2). However, it is unknown whether other genes in MHC region contributes to schizophrenia risk. *GLT8D1* is located in Chromosome 3. Genetic variants near *GLT8D1* showed genome-wide significant association with schizophrenia (1). Nevertheless, this region also contains several other genes, including *PBRM1*, *GNL3*, *GLT8D1*, *SNORD19*, *NEK4*, *ITIH1*, *ITIH3* and *ITIH4*.

Since the MHC region contains numerous highly linked genes, no genes were assigned to the MHC region in the published GWAS of schizophrenia (1). And four genes (*GNL3*, *GLT8D1*, *ITIH1* and *ITIH3*) were assigned to the loci containing *GLT8D1* gene.

Though the PGC2 has reported 108 independent loci associated with schizophrenia (1). However, the reported loci usually span a large chromosomal region and contain many genes. Thus, it is difficult to pinpoint the causal gene (or genes) at the reported loci. In addition, it is not known whether the genetic risk variants contribute to schizophrenia susceptibility through affecting protein function (such as missense and nonsense variants) or regulating gene expression. Our study suggests that *ALMS1*, *GLT8D1* and *CSNK2B* may represent promising candidate genes for the reported loci and it is likely that genetic variants near these genes may confer schizophrenia risk by regulating the expression level of these genes. **Page 32, lines 18-22, Page 33, lines 1-5.**

Or is there any published study that has already linked these two genes with the two GWAS loci?

Response: We performed literature search (using following keywords: (1) *GLT8D1* and Schizophrenia; (2) *CSNK2B* and Schizophrenia) and found that no study has reported the association between *GLT8D1* and schizophrenia. However, a study published in 2014 showed that *ITIH3* polymorphism may confer susceptibility to psychiatric disorders by altering the expression levels of *GLT8D1* (3). No published study showed the association between *CSNK2B* and schizophrenia.

A related question is that the authors used a list of GWAS significant genes. Is this list based on the nearest genes of GWAS loci?

Response: We thank the reviewer for pointing this out. GWAS significant genes were mainly extracted from the largest GWAS of schizophrenia so far (Supplementary Table 3 in the original GWAS paper). In this paper, the authors first defined the genomic regions containing genome-wide significant genetic variants (Due to the linkage disequilibrium, each reported locus usually spans a large chromosomal region and contains several genes). The authors then extracted the genes located within the defined regions. To capture putative regulatory elements, boundaries of each gene are expanded by 20 kb on each side prior to extraction. If the region does not contain a gene (e.g., locus 30), the nearest gene within 500 kb was extracted. In addition, we also extracted genes identified by other genome-wide studies from the SZDB database (4) (<http://www.szdb.org/>). A total of 374 genes were

extracted. **Page 11, lines 20-22; Supplementary Material: Page 5, lines 11-20, Supplementary Material: Page 22, Table S2.**

The authors made statements in quite a few places without statistical justifications:

303: "most of the 10 top candidate genes identified by Sherlock were widely expressed in human brain regions": Should quantify the statistical evidence here.

Response: Following the reviewer's suggestion, we have added the exact number of genes in the revised manuscript. In addition, we also quantified the statistical evidence in the revised manuscript. Page 11, lines 4-16; Page 22, line 22; Page 23, line 1.

323: "frequently interacted with proteins encoded by GWAS significant schizophrenia risk genes". No statistical evidence/p-value.

Response: We appreciate the reviewer for this valuable suggestion. According to the reviewer's suggestion, we provided statistical evidence (i.e., P-value) of the PPI analysis in the revised manuscript. Page 12, lines 2-11, Page 23, lines 18-20.

310: "Permutation test showed that the observed expression pattern is unlikely due to chance effect". What is being tested here? The details of this test should be provided in the text (at least in Methods), not in supplement.

Response: Sherlock integrative analysis (using brain eQTL from Myers et al) identified 10 schizophrenia risk genes. We analyzed the expression pattern of these genes in developing and adult human brain and found that schizophrenia risk genes identified by Sherlock have higher expression level at early developmental stage than later stage (Figure 1). To evaluate if the observed expression pattern is expected by chance or not (i.e., if the genes identified by Sherlock analysis showed higher expression at early developmental stage than later stage), we performed permutation test. Briefly, genes that match the number of Sherlock genes in human genome were randomly selected (8 genes each time, though Sherlock analysis identified 10 genes, 2 of them (LOC376138 and LOC375768) were not found in the BrainSpan dataset) and their expression pattern in developing and adult human brain were assessed. 1000 permutations were performed to obtain the P value (the proportion of tests that have same or extreme expression pattern than the observed expression pattern).

Following the reviewer's suggestion, we provided the details of this test in methods section in the revised manuscript. Page 11, lines 4-16.

161: Co-expression analysis. It is not clear what the authors tested: number of genes co-expressed with known SCZ genes? Or the average level of co-expression? The key details should be provided here.

Response: We are sorry for the confusion. We tested the number of known SCZ genes co-expressed with genes identified by Sherlock integrative analysis. According to the

reviewer's suggestion, we have provided the details of the co-expression analysis in the revised manuscript. **Page 12, lines 18-22, Page 13, lines 1-12.**

Reviewer #3 (Remarks to the Author):

In this study, Yang et al. integrated association signals of a largest GWAS report (PGC2) and brain eQTL data, and carried out a series of statistical analysis. In total, 10 candidate causal genes were predicted, while 3 of them, ALMS1, CSNK2B and GLT8D1 were highlighted by the authors, with validation analyses with different methods or in independent data. The current manuscript was well written, and the analysis is sound.

Response: We thank the reviewer for the positive comment on our work.

I only have the following 2 points:

1. The brain eQTL data mainly used in this study included 193 normal human subjects, which is relatively smaller than a recent publication (Mads et al. Am J Hum Genet. 2017 Jun 1;100(6):885-894.), which integrated GWASs with 24 studies of expression quantitative trait loci. In addition, PGC2 schizophrenia dataset is also one of the studied 57 GWASs in the Mads et al's report. It will be better for the authors to refer to these studies and discuss the advantages and shortcomings of the current study.

Response: We appreciate the reviewer for this valuable suggestion. Following the reviewer's suggestion, we have referred to the study of Mads et al and discussed the advantages and shortcomings of the current study in the revised manuscript. **Page 36, lines 10-22.**

2. I am wondering whether the authors have western blotting results after transfections the RNAi vectors in the cell-based assay, which might provide clearer evidences for the related downstream pathways after knocking-down GLT8D1.

Response: We thank the reviewer for this valuable suggestion. In the revised manuscript (Page 29, lines 1-11, Figure 4 and Supplementary Figure S10), we first validated the RNAi knockdown efficiency targeting to GLT8D1 and CSNK2B by both real-time PCR and western blot assays, and further explored the effects on NSCs self-renew and found that the expression of the well-characterized stemness marker genes of NSCs (such as Klf4, Sox2, Nanog and Nestin) was significantly up-regulated by depleting GLT8D1 and CSNK2B in neural stem cells compared with controls, which could be Wnt signaling pathway dependent (Supplementary Figure S10). However, limited by the lack of transcriptional and proteome data affected by GLT8D1 and CSNK2B knockdown, the detail mechanisms by which GLT8D1 and CSNK2B regulate schizophrenia need to be further verified in future.

Reviewer #4 (Remarks to the Author):

Yang et al performed GWAS studies and identified 10 candidate genes that contribute to schizophrenia risk. Of these 10 candidate genes, the authors identified *ALMS1*, *CSNK2B* and *GLT8D1* as the most promising candidates and validated their expression in the human schizophrenia patient cases. Furthermore, they demonstrated that *GLT8D1* knockdown using shRNA promotes the self-renew and proliferation abilities of cultured neural stem cells (NSCs) and inhibits NSCs differentiation into astrocytes and neurons, providing the evidence of the potential role of identified risk genes in neurodevelopment. Overall, this study identified novel schizophrenia risk genes from GWAS analysis, validated their expression and hippocampal structures in human schizophrenia patient cases.

Response: We thank the reviewer for the positive comment on our work.

However, I have major concerns of the functional studies they performed in this study. The points are listed as below:

1. *The authors performed all of their analysis in human cases, including GWAS, candidate gene expression analysis, and hippocampal structure analysis but the functional studies were performed in the mouse early postnatal (P7) NSC cultures. It is unclear whether human NSCs express these genes and whether the expression pattern of these genes in the mouse hippocampus is similar to that in human.*

Response: We understand the reviewer's concern. We explored the expression of *ALMS1*, *GLT8D1* and *CSNK2B* in human NSCs using expression data from Lafaille et al (5). The Log₂ transformed expression level of *ALMS1*, *GLT8D1* and *CSNK2B* were 7.85, 6.69 and 12.22. We compared the expression level of these three genes with *SOX2*, a gene that is highly expressed in neural stem cells (*SOX2* is a marker for NSCs). The Log₂ transformed expression level of *SOX2* is 12.66. These expression data shows that *CSNK2B* is highly expressed, whereas *ALMS1* and *GLT8D1* are moderately expressed in human NSCs.

Moreover, it is unclear why postnatal mouse NSC culture was used, since schizophrenia was considered as a neurodevelopmental disorder. Mouse embryonic culture appears to be more relevant.

Response: We thank the reviewer for this valuable suggestion. We agree with the reviewer that schizophrenia was considered as a neurodevelopmental disorder. Nevertheless, multiple lines of evidence suggest that dysfunction of hippocampus plays an important role in schizophrenia. We thus used postnatal mouse NSCs in the original manuscript.

Following the reviewer's suggestion, we repeated the experiments using NSCs from the embryonic mouse (E14.5) and observed similar results. Again, we found that knockdown of *GLT8D1* and *CSNK2B* promoted proliferation and inhibited differentiation of NSCs, respectively. We have included these new results in the revised manuscript. **Figure 4, Figure**

5, Page 16, lines 5-22, Page 17, lines 1-22, Page 18, lines 1-11, Page 28, lines 8-22, Page 29, lines 1-22, Page 30, lines 1-3.

To best validate the neurodevelopmental deficits associated with human schizophrenia, patient-derived induced pluripotent stem cell (iPSC) culture should be considered in this context.

Response: We agree with the reviewer that patient-derived induced pluripotent stem cell (iPSC) culture may be useful for validation the neurodevelopmental deficits. However, currently, it is difficult for us to conduct this experiment as it is difficult for us to obtain live cells from schizophrenia patients and we are not familiar with iPSC technology. In addition, although the expression of *ALMS1*, *GLT8D1* and *CSNK2B* was significantly dysregulated in brain tissues of schizophrenia patients in population compared with controls, we are not sure if the expressions of these genes were also dysregulated in NSCs derived from individual schizophrenia cases especially after using iPSC system. Thus, we were not able to perform this assay at current stage.

*2. The authors state that *ALMS1*, *CSNK2B* and *GLT8D1* play active roles in brain development, and showed that reduced *GLT8D1* expression led to deficits in NSC proliferation and differentiation. However, they have not shown whether *ALMS1* and *CSNK2B* exhibit similar NSC phenotypes. Despite that other studies showed the role of *ALMS1* and *CSNK2B* in the central nervous systems, these phenotypes are not relevant to hippocampal structure deficits observed in schizophrenia patient cases.*

Response: We thank the reviewer for this valuable suggestion. According to the reviewer's suggestion, we performed additional experiments in the revised manuscript. Our expression analysis showed that *CSNK2B* was also significantly down-regulated in schizophrenia cases compared with controls (Figure 3C). We thus used shRNA to knockdown *CSNK2B* in embryonic NSCs. As stated above accordingly, knockdown of *GLT8D1* and *CSNK2B* promoted proliferation and inhibited differentiation of NSCs. We have included these new results in the revised manuscript. **Figure 4, Figure 5, Page 16, lines 5-22, Page 17, lines 1-22, Page 18, lines 1-11, Page 28, lines 8-22, Page 29, lines 1-22, Page 30, lines 1-3.**

Compared with *GLT8D1* and *CSNK2B* (which showed significant down-regulation in schizophrenia cases), *ALMS1* was significantly up-regulated in brains of schizophrenia cases (Figure 3A). To mimic the effects of *ALMS1* up-regulation on NSCs, we tried to clone *ALMS1* and construct an over-expression vector. Nevertheless, we found that the coding sequence of *ALMS1* is quite large (over 12 kb), though we tried multiple strategies, even by purchasing the commercial *ALMS1* expression vector, to solve the clone problem, so far we failed. Due to the technical difficulty and limiting time, we thus were not able not investigate the function of *ALMS1* in this study. However, Heydet et al. found that *Alms1* is localized at the base of cilia in hypothalamic neurons and a truncating mutation of *Alms1* reduces the number of hypothalamic neuronal cilia in mice (6). Intriguingly, *ALMS1* is physically interacted with *PCMI* (encoded by a schizophrenia risk gene) (7) and Ceroni et al. (European Society of Human Genetics 2016: Title: P09.045A-Case report: exome sequencing

of a family with childhood disintegrative disorder) (<http://www.well.ox.ac.uk/presentations-and-posters>) found that both of *ALMS1* and *PCMI* genes are mutated in cases of a family with childhood disintegrative disorder (a rare and severe neurodevelopmental disorder). These studies indicated that *ALMS1* plays a vital role in brain development and dysfunction of *ALSMI* causes neurodevelopmental disorder.

3. The authors showed *GLT8D1* knockdown in mouse NSCs led to increased self-renewal, and decreased neuronal and glial differentiation. The authors used *GFAP* marker for astrocytes. However, in the NSC culture, it has been widely accepted that NSCs give rise to oligodendrocytes besides neurons and astrocytes. This oligo marker should be quantified as well.

Response: We thank the reviewer for this valuable suggestion. Following the reviewer's suggestion, we carried out additional experiments through quantifying the oligo marker (O4). We found knockdown of *GLT8D1* and *CSNK2B* significantly impaired the differentiation of NSCs into oligodendrocytes. We included these new results in the revised manuscript. **Figure 5C, D, Page 17, lines 18-22, Page 18, lines 1-11, Page 29, lines 12-22.**

4. Schizophrenia is considered as a synaptic disorder characterized by aberrant synapses (both excitatory and inhibitory). Dendritic and synaptic phenotypes should be included in the analysis associated with *GLT8D1* knockdown, including dendritic length, branches, crossings, and functional synaptic characterizations (mIPSCs and mEPSCs).

Response: We appreciate the reviewer for this valuable suggestion. According to the reviewer's suggestion, we performed additional experiments to address these questions. We found that knockdown of *GLT8D1* and *CSNK2B* significantly affected the dendritic length and branches of neurons. In addition to the dendritic phenotypes, synaptic transmission was also affected in *GLT8D1* and *CSNK2B* knockdown neurons. These results suggest that the risk genes identified in this study (i.e., *GLT8D1* and *CSNK2B*) may confer risk of schizophrenia through affecting neurodevelopment and synaptic transmission. We included these new results in the revised manuscript. **Figure 6, Figure 7, Page 18, lines 19-22, Page 19, lines 1-15, Page 30, lines 5-20.**

Again, we really appreciate the reviewers for their insightful and valuable comments and suggestions, which help us to improve the quality of the manuscript significantly. We hope that our revised work is satisfactory, and are happy to further improve it if needed.

References

1. Schizophrenia Working Group of the Psychiatric Genomics Consortium* (2014) Biological insights from 108 schizophrenia-associated genetic loci. *Nature*, **511**, 421-427.
2. Sekar, A., Bialas, A.R., de Rivera, H., Davis, A., Hammond, T.R., Kamitaki, N., Tooley, K., Presumey, J., Baum, M., Van Doren, V. *et al.* (2016) Schizophrenia risk from complex variation of complement component 4. *Nature*, **530**, 177-183.
3. Sasayama, D., Hori, H., Yamamoto, N., Nakamura, S., Teraishi, T., Tatsumi, M., Hattori, K., Ota, M., Higuchi, T. and Kunugi, H. (2014) ITIH3 polymorphism may confer susceptibility to psychiatric disorders by altering the expression levels of GLT8D1. *J Psychiatr Res*, **50**, 79-83.
4. Wu, Y., Yao, Y.G. and Luo, X.J. (2017) SZDB: A Database for Schizophrenia Genetic Research. *Schizophr Bull*, **43**, 458-471.
5. Lafaille, F.G., Pessach, I.M., Zhang, S.Y., Ciancanelli, M.J., Herman, M., Abhyankar, A., Ying, S.W., Keros, S., Goldstein, P.A., Mostoslavsky, G. *et al.* (2012) Impaired intrinsic immunity to HSV-1 in human iPSC-derived TLR3-deficient CNS cells. *Nature*, **491**, 769-773.
6. Heydet, D., Chen, L.X., Larter, C.Z., Inglis, C., Silverman, M.A., Farrell, G.C. and Leroux, M.R. (2013) A truncating mutation of *Alms1* reduces the number of hypothalamic neuronal cilia in obese mice. *Dev Neurobiol*, **73**, 1-13.
7. Gurling, H.M., Critchley, H., Datta, S.R., McQuillin, A., Blaveri, E., Thirumalai, S., Pimm, J., Krasucki, R., Kalsi, G., Queded, D. *et al.* (2006) Genetic association and brain morphology studies and the chromosome 8p22 pericentriolar material 1 (PCM1) gene in susceptibility to schizophrenia. *Arch Gen Psychiatry*, **63**, 844-854.

Reviewer #1 (Remarks to the Author):

NO more comments. I approve this version.

Reviewer #2 (Remarks to the Author):

The authors have addressed all my comments. I recommend it for publication.

Reviewer #3 (Remarks to the Author):

My original comments have been addressed by the authors.

A new paper (Nature Genetics 49, 1576–1583 (2017)) for a large-scale GWAS of Chinese individuals and a trans-ancestry meta-analysis with PGC2 data was published during the submission. The findings from the newest paper should be referred and discussed.

Reviewer #4 (Remarks to the Author):

The authors have addressed my comments with additional experiments. Now the manuscript is ready for publication.

REVIEWERS' COMMENTS:

We thank the third reviewer for additional comments. We have responded to his/her comments point-by-point as below (in red) and have marked all the relevant changes in the revised manuscript.

Reviewer #1 (Remarks to the Author):

NO more comments. I approve this version.

Response: We thank the reviewer for his approval of our manuscript.

Reviewer #2 (Remarks to the Author):

The authors have addressed all my comments. I recommend it for publication.

Response: We thank the reviewer for his recommendation for publication of our manuscript.

Reviewer #3 (Remarks to the Author):

My original comments have been addressed by the authors.

A new paper (Nature Genetics 49, 1576–1583 (2017)) for a large-scale GWAS of Chinese individuals and a trans-ancestry meta-analysis with PGC2 data was published during the submission. The findings from the newest paper should be referred and discussed.

Response: We thank the reviewer for this suggestion. Following the reviewer's suggestion, we have referred and discussed the findings from the newest paper. Intriguingly, we found that eSNPs of *GLT8D1* and *CSNK2B* have a smaller *P* values (*P* value of rs6795646 is 4.56×10^{-10} , and *P* value of rs2075800 is 1.56×10^{-13}) in the paper mentioned by the reviewer (i.e., transancestry GWAS meta-analysis, PGC2-plus-Chinese, 43,175 cases and 65,166 controls) compared with PGC2, indicating that these two SNPs have same risk alleles in Chinese population. These data provide further support for the involvement of *GLT8D1* and *CSNK2B* in schizophrenia. Page 36, lines 19-22, Page 37, lines 1-6.

Reviewer #4 (Remarks to the Author):

The authors have addressed my comments with additional experiments. Now the manuscript is ready for publication.

Response: We thank the reviewer for his recommendation for publication of our manuscript.